# Flexible temperature-pressure dual sensor based on 3D spiral thermoelectric Bi$_2$Te$_3$ films

Hailong Yu[1,2], Zhenqing Hu[1,2], Juan He[1,2], Yijun Ran[1,2], Yang Zhao[2], Zhi Yu [1,2] ✉ & Kaiping Tai [1,2,3] ✉

Dual-parameter pressure-temperature sensors are widely employed in personal health monitoring and robots to detect external signals. Herein, we develop a flexible composite dual-parameter pressure-temperature sensor based on three-dimensional (3D) spiral thermoelectric Bi$_2$Te$_3$ films. The film has a *(000l)* texture and good flexibility, exhibiting a maximum Seebeck coefficient of −181 μV K$^{-1}$ and piezoresistance gauge factor of approximately −9.2. The device demonstrates a record-high temperature-sensing performance with a high sensing sensitivity (−426.4 μV K$^{-1}$) and rapid response time (~0.95 s), which are better than those observed in most previous studies. In addition, owing to the piezoresistive effect in the Bi$_2$Te$_3$ film, the 3D-spiral deviceexhibits significant pressure-response properties with a pressure-sensing sensitivity of 120 Pa$^{-1}$. This innovative approach achieves high-performance dual-parameter sensing using one kind of material with high flexibility, providing insight into the design and fabrication of many applications, such as e-skin.

Recent advances in flexible electronics and materials have led to significant progress in wearable technologies, including implantable devices[1–3], electronic skin[4,5] (e-skin), and physiological signal monitoring[6,7]. These developments offer potential for the long-term continuous monitoring of human activities and health conditions in daily life. To realize these applications, it is vital to develop highly sensitive and cost-effective multifunctional flexible sensors. Among the various types of flexible sensors, on-skin electronic sensors that can accurately differentiate between temperature and pressure stimuli are of significance for health monitoring systems.

Most active temperature sensors are based on thermal resistance effects[8] and are driven by external batteries that require frequent charging and replacement, making long-term unattended monitoring challenging[9]. Therefore, the development of passive temperature sensors is desirable. Thermoelectric (TE) generation is an attractive technology that directly converts heat into electrical voltage[10], and the voltage intensity is directly correlated with the changes in temperature difference. Thus, TE devices can achieve passive temperature sensing by monitoring changes in voltage signals without an external power supply. Temperature sensors based on the TE effect, such as thermocouples, have been developed, however, commercial thermocouples are not suitable for flexible applications. To achieve a highly flexible body attachment, most flexible temperature sensors employ organic TE materials as the active components. However, their lower Seebeck coefficients limit the device performance. For example, Kyung and Heesuk[11] employed carbon-nanotube-yarn to create a flexible TE generator with a Seebeck coefficient of <80 μV K$^{-1}$. Similarly, Zhang and Bae[12] developed a temperature sensor with a sensitivity of 11 μV K$^{-1}$ using poly(3,4-ethylenedioxythiophene):poly(styrene sulfonate) (PEDOT:PSS) and Ag nanoparticle hybrid ink. However, high-performance inorganic TE materials are significantly limited in flexible sensor applications due to their native rigidity and brittleness[13].

[1]School of Materials Science and Engineering, University of Science and Technology of China, Shenyang 110016, China. [2]Shenyang National Laboratory for Materials Science, Institute of Metal Research, Chinese Academy of Sciences, Shenyang 110016, China. [3]Liaoning professional technology innovation center for integrated circuit thermal management, Shenyang 110016, China. ✉e-mail: zyu@imr.ac.cn; kptai@imr.ac.cn

This issue can be solved by fabricating inorganic TE materials assembled on a thin flexible substrate[14] to fabricate flexible thin-film TE devices. However, the lateral structure design of most thin-film TE devices[15] resulting in a long response time owing to the long and thin TE leg as a heat and current diffusing route[16], reducing sensing performance. As reported by Huixu[17] and Ya[18], where their length of the device is 10 mm and 6 mm, their response time is 34 s and 17 s. Typically, for planar TE devices (except for microdevices), the length of the thermoelectric leg is in the millimeter range. The thickness of the film is typically only in the range of a few hundred nanometers to several tens of micrometers. Therefore, adopting a vertical structure can significantly reduce the response time. Yingming[19] used a SnSe film with 366 nm thick to realize an ultrafast infrared detection with a response time of 11.3 microseconds. However, vertical thin-film devices face a significant challenge in creating a sufficient and stable temperature difference in the out-of-plane direction owing to its low thermal impedance. Studies have attempted to adjust the out-of-plane temperature difference by creating thin-film TE devices with three-dimensional (3D) structures to maximize thermal impedance. For example, Rogers[20] created a 3D helical structure to achieve efficient thermal impedance matching. Guo[21] fabricated a kirigami-based structure to obtain the effective temperature difference. However, the 3D hybrid structure of thin-film TE devices has barely been investigated, owing to the complexity of fabricating a 3D structure.

Pressure sensors based on various physical effects, such as piezoresistive[22,23], piezocapacitive[24,25], piezoelectric[26,27], and triboelectric[28,29] effects, have been widely employed. Among them, piezoresistive pressure sensors have been investigated because of their simple structure and easy read-out mechanism[30]. Piezoresistive materials demonstrate a change in their electrical resistance in response to applied mechanical stress or pressure, which is attributed to the stress-induced changes in the band structure[31]. Thus, one can easily infer the pressure on the device by the piezoresistive effect via measuring the resistance change of the sensor under an external pressure stimulus. Currently, most studies on pressure/temperature sensors require the integration of two types of materials with temperature and pressure–response properties to achieve multifunctional sensing[16,32]. However, complex preparation processes and difficulties in increasing the integration density owing to limited space significantly restrict sensor's practical applications. Bismuth telluride ($Bi_2Te_3$), exhibiting the best TE performance near room temperature, also exhibits piezoresistive effects, making it possible to achieve the dual-functional sensing of temperature and pressure using one material. However, the Seebeck coefficient of $Bi_2Te_3$ will be changed under the strain, which is studied by Weiliang[33] and Hajji[34]. They used first principles to study the effect of thermoelectric performance under the biaxial mechanical strains (pressure and tensile). They both found that the Seebeck coefficient increases under compressive strain and decreases under tensile strain. These theoretical results remind us of the need to pay attention to the effect of strain on the Seebeck coefficient and to correct it as much as possible when using $Bi_2Te_3$ for temperature and pressure sensing.

Owing to its excellent properties, $Bi_2Te_3$ films exhibit significant potential as a highly suitable active material in multifunctional pressure and temperature sensors. Herein, we report a temperature–pressure dual-functional sensor based on a (000l)-textured $Bi_2Te_3$ film deposited on a polyimide (PI) substrate. The $Bi_2Te_3$ exhibits a high Seebeck coefficient of −179 μV K$^{-1}$ with electrical conductivity of ~700 S cm$^{-1}$, thus the power factor reached 22.6 μW cm$^{-1}$ K$^{-2}$ at room temperature. Meanwhile, its piezoresistance gauge factor reached ~−9.2 due to the highly (000l) texture. These outstanding performances allowed our sensor to detect temperature and pressure signals using TE and piezoelectric effects. A 3D-spiral structure was prepared using a lab-built femtosecond laser and device-integrated equipment (Supplementary Fig. 15) to match the out-of-plane thermal impedance requirement, such as temperature-sensing

e-skins. Furthermore, based on the piezoresistive effect, the resistance fluctuation of the device caused by the compression deformation of the spiral structures were employed to sense the external pressure. Temperature–pressure dual-parameter sensing is thus achieved by monitoring changes in voltage and resistance. Finally, we developed 3 × 3 bimodal sensor arrays to detect the temperature and pressure maps, which is a promising route for future applications.

## Results

### Thermoelectric and piezoresistive effect of the flexible $Bi_2Te_3$ film

The TE performance of the $Bi_2Te_3$ film was optimized and the details are provided in Supplementary Note 1 and Supplementary Figs. 1–4. Figure 1a shows the TE performance of the optimized $Bi_2Te_3$ film. It is well known that the deposition temperature affects the crystalline state[35,36], directly influencing TE properties. In this study, a high deposition temperature of 613 K is implemented and a high crystal quality $Bi_2Te_3$ film with a (000l) texture was achieved to improve the electrical conductivity, which can be identified in the scanning electron microscopy image in Supplementary Fig. 1c, the Electron Back-Scattered Diffraction (EBSD) image in Supplementary Fig. 2 and the X-ray diffraction (XRD) pattern in and Supplementary Fig. 4. Due to the different linear thermal expansion coefficients (LTECs) between $Bi_2Te_3$ ($17.11 \times 10^{-6}$ K$^{-1}$)[37] and polyimide ($20 \times 10^{-6}$ K$^{-1}$), the in-plane compressive stresses created inside the film. So, the (006) and (015) peaks shift towards the lower angle with the temperature increase in the spectrum. The electrical conductivity decreased from 702 to 635 S cm$^{-1}$ with increasing temperature (Fig. 1a), indicating that the $Bi_2Te_3$ film exhibited metallic-like or degenerate semiconductor transport behavior, and the interrelationship between the carrier concentration and Seebeck coefficient can thus be described by single parabolic band models of electron transport[38]. The low carrier concentration (Table S1, $n_e = -4.54 \times 10^{19}$ cm$^{-3}$) resulted in a high Seebeck coefficient, $|S|$, reaching a maximum of approximately 181 μV K$^{-1}$ (Fig. 1a). The power factor was thus determined to be 22.6 μW cm$^{-1}$ K$^{-2}$, which is a competitive value compared to the literature[39] for $Bi_2Te_3$ films.

The (000l)-textured $Bi_2Te_3$ film was bonded to a polyethylene terephthalate substrate to investigate the piezoresistive effect by calculating the gauge factor (GF), which is defined[31] as $GF = \varepsilon \triangle R / R_0$, where $\varepsilon$ denotes the strain, $\triangle R$ is the resistance change under the strain, and $R_0$ is the initial resistance of the film. The strain was applied to the $Bi_2Te_3$ film by controlling its bending (up or down) through a motorized translation stage, and the bending radius was finely controlled by tuning the moving steps using the bend test system shown in Supplementary Fig. 17. Figure 1b shows the time-dependent evolution of the relative resistance ($\triangle R/R_0$) of the $Bi_2Te_3$ film in response to uniaxial strain, as the bending radius is gradually increased and then decreased. The curve displays numerous steps, each representing a bending radius, which is identified through camera image recognition. The $\Delta R/R_0$ presents a stepwise decrease with the increase of stepwise bending radius under positive curvature, while presents a stepwise increase with the decrease of stepwise bending radius under negative curvature. The bend radii are 6.3 mm, 4.7 mm, 4 mm, 3.4 mm, and 2.9 mm, respectively. In the case of negative curvature bending, the resistance change exhibits the opposite behavior. The bend radii are 5.4 mm, 4.9 mm, 4 mm, 3.4 mm, and 3.1 mm, respectively. So, an obvious piezoresistive phenomenon is observed as the $\triangle R/R_0$ decreases under a positive bend curvature ($\rho > 0$), whereas $\triangle R/R_0$ increases under a negative curvature ($\rho < 0$). The strain in the film is estimated by $\varepsilon = d/2r$, where $d$ and $r$ denote the thickness and curvature radius of the PI film, respectively[40]; $\triangle R/R_0$ of the $Bi_2Te_3$ film is plotted as a function of the strain in Fig. 1c, demonstrating a negative piezoresistance and linear change with strain. Thus, the GF was calculated to be approximately −9.2, which is ~8 times larger than the recently reported values[41] and better than most metal materials[31],

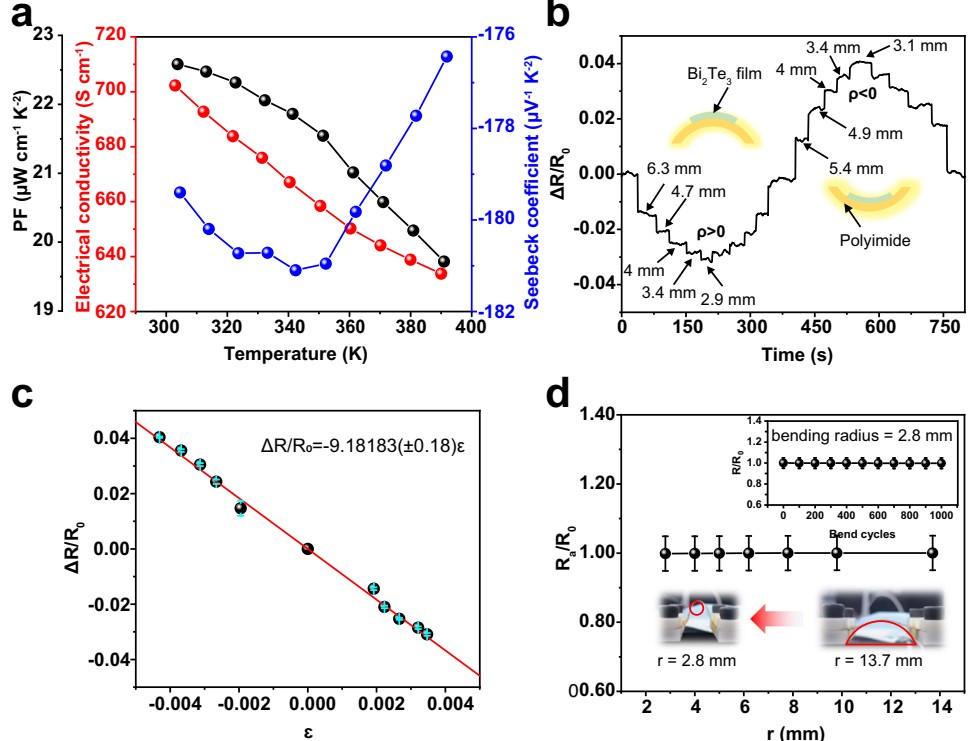

**Fig. 1 | Electric and flexible properties of Bi$_2$Te$_3$/PI (polyimide) films deposited at 613 K. a** Temperature-dependent Seebeck coefficient, electrical conductivity, and power factor of the film. **b** Real-time piezoresistance response curve of the film. The numbers indicated by the arrows represent the bending radius. **c** Relative resistance changes of the film as a function of strain. The red line is the fitted curve, whose slope denotes the gauge factor of −9.2. Error bars represent the standard deviation for resistance. **d** Relative electrical resistance as a function of bending radius for the film. $R_a$ and $R_0$ denote the resistance of the film after bending deformation and the original flat state, respectively. The red arrow symbolizes a transition from a larger to a smaller bending radius. Inset: results of the cyclic bending test under $r = 2.8$ mm for the film. Error bars represent the measurement uncertainties for resistance from instrument (-5%). Source data are provided as a Source Data file.

demonstrating its applicability in strain and pressure sensing applications.

Furthermore, the bending flexibility of the (000l)-textured Bi$_2$Te$_3$ film was investigated. Referring to the original resistance, $R_O$, the normalized resistance ($R_a/R_O$) after a single bend is plotted in Fig. 1d as a function of the bend radius. The resistance remains nearly unchanged even after a bend radius of 2.8 mm. The inset in Fig. 1d demonstrates that the resistance remained stable after 1000 bending cycles at 2.8 mm, indicating the excellent stability of the film, significantly better than previous reports[13,14]. Strain analysis and the flexible test are discussed in Supplementary Note 2, and the result is shown in Supplementary Fig. 5. The thickness of the film is an important factor in achieving flexibility. As can be seen in Supplementary Fig. 5a, the coefficient $\frac{1+2\eta+\chi\eta^2}{1+\chi\eta}$ decreases with the $d_{sample}/d_{subtrate}$ increase and then increase with the $d_{sample}/d_{subtrate}$ increase. Until $d_{sample}/d_{subtrate}$ reaches approximately 0.16, coefficient $\frac{1+2\eta+\chi\eta^2}{1+\chi\eta}$ reached the minimum. It means if the thickness of polyimide is 25 μm, the $\varepsilon_b$ will reach the minimum value when the $d_{sample} \approx 4$ μm, which can effectively improve the flexibility. Cause the thickness of the films we deposited is less than 4 μm, the flexibility will be better with the thickness increase theoretically. Furthermore, the thinner substrate exhibits a lower bending strain, resulting in better flexibility[42]. Supplementary Fig. 5b and c show the stress nephogram of the film under different bend radii and substrate thicknesses. It is easy to know the strain where $d_{substrate} = 125$ μm is much larger than where $d_{substrate} = 25$ μm under the same bend situation and $d_{sample}$. As can be seen, when the $d_{sample}$ is close to 0, the strain in the film is about 3% at $d_{substrate} = 125$ μm under a bend radius of about 2 mm, which is 5 times larger than strain ($\approx 0.6\%$) at $d_{substrate} = 25$ μm. Moreover, the layered structure of Bi$_2$Te$_3$ with a

highly textured (000l) orientation is beneficial for flexibility because interlayer slipping reduces stress concentration[43]. As Supplementary Fig. 5d and e shows, the (015)-textured film with 125 μm-thick substrate's resistance doubled after a single bend at the radius of 2.8 mm and increased to 75 times larger than the initial state after 1000 bend cycles. While the (000 l)-textured film with 125 μm-thick substrate's resistance increased only about 5 times after 1000 bend cycles. So, (000 l) texture is indeed conducive to improving the flexibility of Bi$_2$Te$_3$ films. Thus, the Bi$_2$Te$_3$ film with a substrate thickness of 25 μm and a highly textured (000l) microstructure exhibits excellent flexibility.

## Device fabrication and sensing mechanism

Figure 2a, b exhibit a schematic diagram of the spiral structure of a 3-couples-sensor, providing structural flexibility, which is beneficial for the flexibility of the device. The sensor exhibited remarkable flexibility, as evidenced by its ability to withstand bending of over 90° using fingers, as shown in Fig. 2c. The 3D-spiral structure flexible pressure–temperature sensor contains three components. The sensing active component was Bi$_2$Te$_3$ film deposited on a 25-μm-thick PI layer; the electrode component was 4.5-μm-thick Cu foils and deposited Au film; the encapsulation component was poly(dimethyl siloxane) (PDMS). PDMS can be divided into three layers: the top encapsulation, intermediate filling, and bottom supporting layers (Supplementary Note 3 and Supplementary Fig. 7). The structure features of a sensor and the electrical interconnection are shown in Fig. 2d and Supplementary Fig. 6. As can be seen in Fig. 2d, the sensor contains 3-pair legs. Figure 2e shows the 3D structure of the spiral constructed by X-ray tomography (XRT)[44], which shows an inverted tower-type

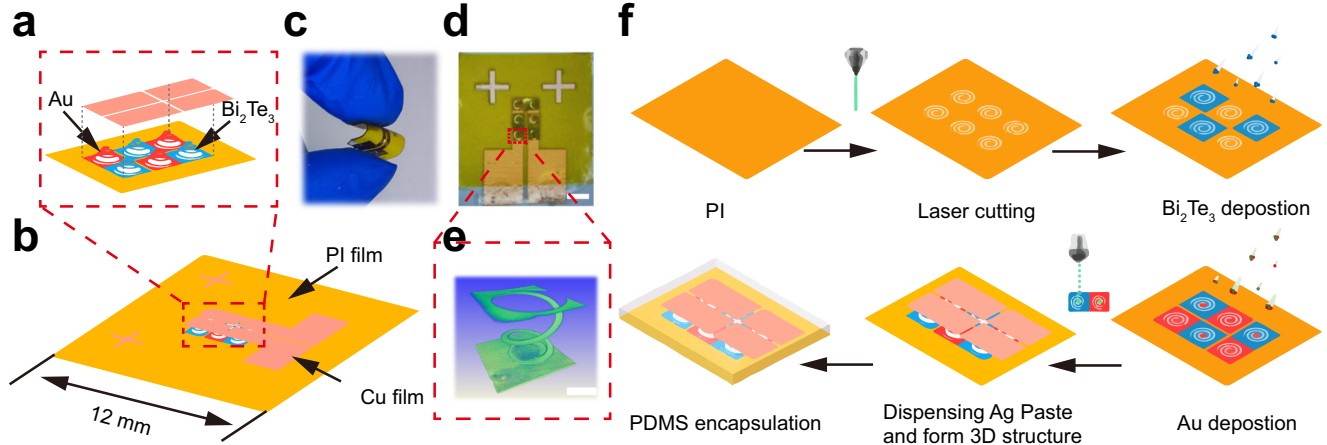

**Fig. 2 | Temperature and pressure sensors along with an illustrative diagram of the manufacturing process. a** Magnified image of the part shows the 3D (three-dimensional) spiral structures. **b** Schematic of the sensor. PI: polyimide. **c** Sensor folding by hand, demonstrating its excellent flexibility. **d** Optical photograph of the 3-pair leg device. Scale bars, 2 mm. **e** A 3D-spiral structure of one leg in a device constructed by XRT. Scale bars, 250 μm. **f** Schematic for the fabrication of the 3D

pressure–temperature sensor. First, the prepared PI film was cut to 2D spiral structure. Subsequently, $Bi_2Te_3$ and Au film were deposited on the spiral structure by magnetron sputtering. The 3D-spiral structure was then constructed using dispensing and pulling process. Finally, PDMS (Polydimethylsiloxane) applied for sensor encapsulation.

spiral structure. Supplementary Fig. 6a further shows the corresponding cross-section of the spiral structure image of Fig. 2e by XRT, where $Bi_2Te_3$ film, Ag paste, and Cu foil can be seen. It is worth noting that the Ag paste is tightly attached to the Cu foil, which indicates a good electrical connection. Supplementary Fig. 6b, c shows the diagram of the sensor electrical interconnection. Supplementary Fig. 6c depicts a simplified schematic diagram of the Supplementary Fig. 6b. The electrical interconnection of a sensor constructed by three sets of π-type structures connected in series, with one set of π-type structures consisting of two copper foils (black lines) and a pair of interconnected spirals (red and blue spiral). A simplified fabrication process is illustrated in Fig. 2f. First, a 2D spiral pattern of PI was constructed by femtosecond laser processing, followed by the deposition of $Bi_2Te_3$ and Au films through mask magnetron sputtering. A conductive silver adhesive was then employed to connect the central site of the spiral-active materials to the Cu foil electrodes using a dispensing technique. After curing, the 2D spiral patterned PI was pulled to form a 3D-spiral structure using a laboratory-built integrated device. Finally, the PDMS was poured into the structure to encapsulate the device.

During temperature sensing, when a temperature gradient is applied to the upper and lower surfaces of a device, a TE potential is generated between the two sides with an output voltage ($V_T$) signal. $V_T$ is defined as $V_T = N[(S_p − S_n)]A\Delta T = A\alpha\Delta T$, where $S_p$ and $S_n$ denote the Seebeck coefficients of the Au ($S_p = 0$) and $Bi_2Te_3$ films, respectively, $A$ is an effective temperature difference coefficient, $N$ is the number of $p$-$n$ couples, and $\triangle T$ is the temperature difference over the two ends of the device. We build a simple one-dimensional heat conduction model as shown in Supplementary Note 4 and Supplementary Fig. 17. so, the temperature-sensing sensitivity of the sensor is defined according to the formula:

$$S = A\alpha = \left(1 - \frac{R_{s1} + R_{01}}{R_{th} + R_{s1} + R_{01}}\right)\alpha \tag{1}$$

where $R_{th}$ denotes the effective thermal resistance, $R_{01}$ and $R_{s1}$ are the parasitic thermal resistances of the device originating from the PDMS layers, and $S$ is the actual sensitivity of the sensor.

For pressure sensing, the resistance signal was monitored to detect the change in the external pressure based on the piezo-resistance effect of the (000l)-textured $Bi_2Te_3$ film. Thus, when an external pressure is applied to the sensor, compression deformation of

the spiral-structured $Bi_2Te_3$ film occurs, resulting in a change in the resistance of the sensor. Because of the novel 3D-spiral structure and complex contact in the sensor, the strain in the $Bi_2Te_3$ film is complicated; however, we can still form a relationship between the pressure and resistance change according to the experiment. Herein, the pressure, $P$, is defined as $P = K \times \triangle R/R_0 + B$, where $\triangle R$ denotes the resistance change under pressure, $R_0$ the initial resistance of the device, and $K$ (sensitivity of the pressure sensor) and $B$ are coefficients depending on the device structure.

## Device performance measurement

The key parameters of our temperature sensor, such as sensitivity, response time, stability, and resolution, were investigated. Figure 3a shows the sensing performance with five on/off cycles for each $\Delta T$. The on/off switch is achieved by rapidly pressing a hot copper block against and away from the sensor (The input signal of $\Delta T$ and the response time of the sensor under $\Delta T$ can be seen in Supplementary Fig. 12). It can be observed that the output voltage decreases with increasing $\Delta T$ from 0.9 to 17.1 K, and the five cycles of each group exhibit a stable voltage output, demonstrating good sensing repeatability. As shown in Fig. 3b, when a stable temperature difference is applied at the two sides of a 0.4-mm-thick sensor with three $p$-$n$ couples, a Seebeck voltage is generated. The generated voltage decreases with increasing $\Delta T$, exhibiting a linear relationship with a sensitivity of -369.6 μV K$^{-1}$. This value is approximately 3–4 times higher than previous studies[16]. The inset in Fig. 3b shows a magnified view of one peak in the curve from the second picture of Fig. 3a, demonstrating an instant response time of 0.50 s and a rapid recovery time of 0.73 s at $\Delta T = 2.9$ K, outperforming the TE-based temperature sensor in planar structures[40–42].

According to Eq. (1), S is positively correlated with $R_{th}$ and negatively correlated with $R_{01}$ and $R_{s1}$. Thus, $S$ can be further optimized by increasing the height of the thermoelectric spiral or decreasing the thicknesses of the upper and lower PDMS layers because the thermal resistance is proportional to the height or thickness. Here, we increased the height of the spiral structure by controlled stretching to increase $R_{th}$ while maintaining the PDMS thickness with a fixed $R_{01}$ and $R_{s1}$, resulting in a total device thickness of 1 mm. Consequently, the sensitivity of the device was determined to be -426.4 μV K$^{-1}$, as shown in Supplementary Fig. 13a, which is among the highest levels reported in recent studies on flexible thermoelectric temperature sensors. Supplementary Fig. 13c shows the sensor's voltage response to

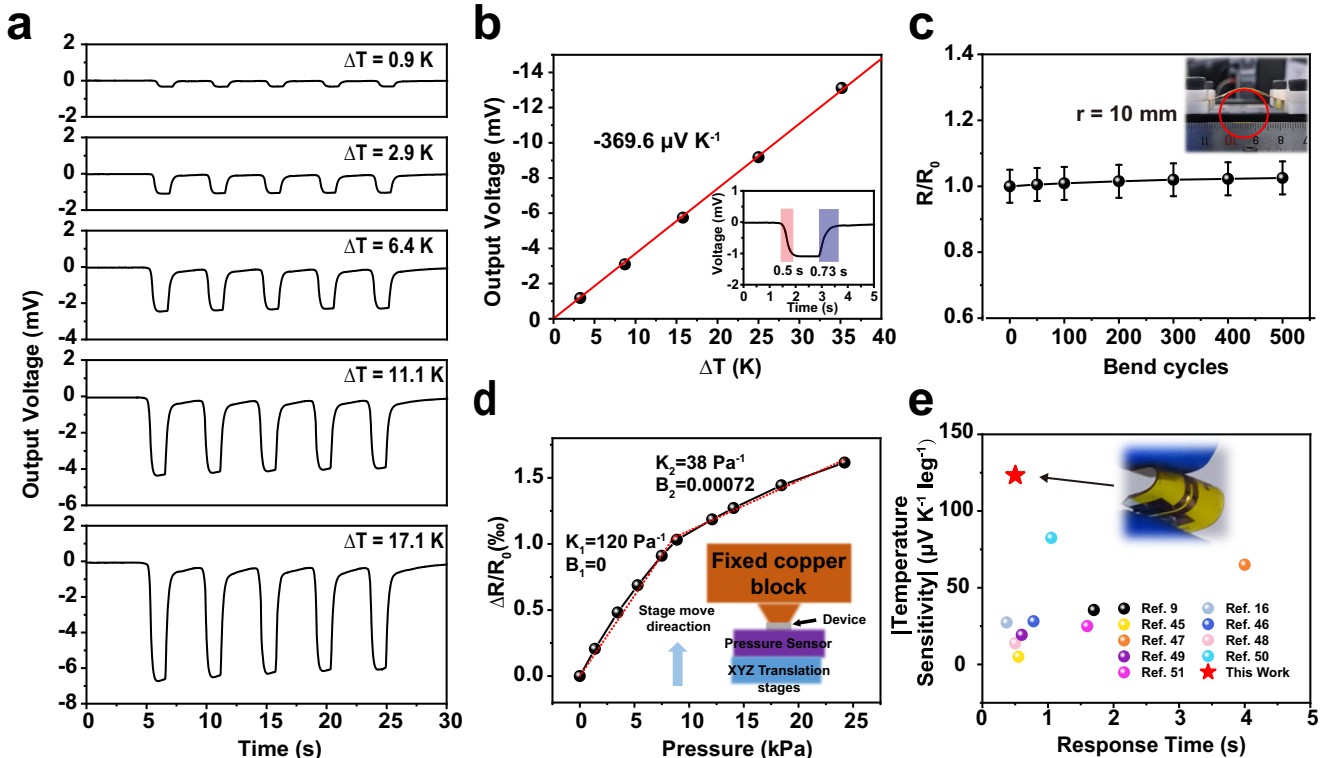

**Fig. 3 | Sensing performance of the pressure–temperature sensor. a** Voltage response to temperature difference stimuli from 0.9 to 17.1 K. **b** Output voltage as a function of the sensor temperature gradient. The red line is the fitted curve, exhibiting a sensitivity of $-369.6\,\mu V\,K^{-1}$. The R-squared value for the linear fit is 0.99984. The inset shows the time-resolved response of the sensor to temperature stimuli with the red and violet zones corresponding to the response and relaxation time, respectively. **c** Resistance variation of the sensor with bending times. Error bars represent the measurement uncertainties for resistance from instrument (-5%). **d** Relative resistance changes as a function of the pressure that is applied to the sensor. The R-squared value for the linear fit in the low-pressure region is 0.9965, while for the high-pressure region, the R-squared value for the linear fit is 0.98968. The inserted image illustrated the test situation. **e** Temperature sensor performance in this study compared to the literature[9,16,45–51]. Source data are provided as a Source Data file.

temperature difference stimuli from 9.1 K to 85.0 K. Furthermore, the device exhibits an instant response time of 0.95 s and a recovery time of 6.37 s at $\Delta T = 20.3$ K (Supplementary Fig. 13b). Those results indicate that there is still room for improving sensor performance by adjusting the sensor structures.

In addition, the flexibility was investigated, as shown in Fig. 3c, by monitoring the resistance change of the device under bending conditions. At a bending radius of 10 mm, after 500 cycles, the resistance of the thermoelectric module demonstrated an increase of <3%, indicating excellent flexibility and the possibility of applications in wearable electronics. The temperature-sensing resolution was further evaluated, using the measurement device shown in Supplementary Fig. 10a. A sensor is attached to the surface of a Thermo Electric Cooler (TEC) covered with embedded thermistor silicone, and the up surface of the sensor is in contact with a custom-made copper block, which temperature is controlled by a PID controller. The pressure sensor is installed under the water-cooling system which is to cool the TEC hot surface when it works. By keeping the copper block temperature constant at 303 K and adjusting the TEC current, a temperature difference is created between the upper and lower surfaces of the device. Supplementary Fig. 10b shows the sensitivity of the sensor under different pressures. The detail is discussed in Supplementary Note 4. As shown in Supplementary Fig. 10c, a temperature gradient of 0.1 K can be observed with steps shown in the V-t curve, indicating that the resolution can reach 0.1 K at room temperature.

We compare the temperature sensing performance of our sensors with the recent reports[9,16,45–51], as shown in Fig. 3e. Most of the previously reported temperature–pressure dual-parameter sensors employ distinct materials to achieve separate sensing functions. In

comparison, the pressure and temperature sensing performance are simultaneously realized in our device by using a single flexible bismuth telluride material. Although many devices with fast response times have been reported in recent years, their improvements in normalized sensitivity remain limited. Our sensor, exhibits a temperature response times less than 0.5 s, and an absolute value of normalized temperature sensitivity of 123.2 $\mu V\,K^{-1}\,leg^{-1}$, demonstrates superior performance compared to the literature reports, as shown in Fig. 3e.

For the pressure-sensing function, we investigated the relationship between the relative alternating current resistance changes at various pressures (Fig. 3d). A three-axis translation stage was employed to control the pressure applied to the devices and the pressure was monitored using a commercial pressure sensor. Consequently, the relative resistance increased with increasing pressure, indicating that the $Bi_2Te_3$ film was under compressive strain because of the negative GF of the film. According to the experimental results, the pressure sensing process undergoes two stages, the low-pressure region from 0–10 kPa with a sensitivity of $K_1 = 120$ Pa$^{-1}$ and a higher pressure region with a sensitivity of $K_2 = 38$ Pa$^{-1}$. This may be owing to the complex stress state in the 3D-spiral structure, causing the relationship between the pressure and relative resistance change to deviate linearly. However, this study demonstrated the possibility of using $Bi_2Te_3$ films with 3D-spiral structures as pressure sensor units in the future. Further, we have tested the pressure-response time of the sensor. Supplementary Fig. 11 shows the resistance change under 3 times 5.3 kPa load and unload process. The curve shows a stable response for the pressure sense and exhibits an instant response time of ~0.3 s and a recovery time of ~1.1 s.

## Practical applications for the temperature–pressure sensors

The temperature-sensing performance under different practical conditions was investigated to determine the feasibility of practical applications. Figure 4a shows the voltage signal generated during the exhalation of an adult male. It can be observed that the sensor will generate a pulse signal corresponding to the blowing frequency. Notably, a negative signal pulse occurred after the recovery of the TE signal. This can be attributed to the liquefied vapor from the mouth evaporating and removing heat from the upper surface of the sensor. This process causes a transient lower temperature of the upper surface compared to that of the lower surface, resulting in a negative voltage signal. As shown in Fig. 4b, when a man's index finger with a temperature of approximately 33 °C touched the device at a room temperature of approximately 31 °C, the output voltage was instantly detected, and the signal remained nearly unchanged during five cycles of touching. Furthermore, the functions of the devices working as thermal switches were investigated. Figure 4c shows the cold/hot-sensing switching system. The sensor was connected to a signal amplifier with two light-emitting diodes (LEDs, red and blue) connected in normal and reverse to the system, respectively. When the finger touches the sensor, the red LED is turned on, indicating a relatively higher temperature. In the middle of Fig. 4c, when the pen touched the sensor, there was no response from the diodes. However, when an ice block is attached to the device, the blue LED is turned on, indicating a low temperature. The sensor also functions as a multistage temperature switch by generating different voltages under various temperature gradients. The generated sensor signal is amplified by a signal amplifier and detected by a single-chip microcomputer, which can further control the number of lighted LEDs by monitoring the voltage changes. As shown in Fig. 4d, when the 37 °C heating pad controlled by the Proportional Integral Derivative (PID) controller touches the device, two LEDs are on, whereas the 50 and 70 °C heaters corresponded to 4 and 6 LEDs, respectively.

For real applications, real-time spatial distributed pressure/temperature sensing on a large scale is necessary, and the integration of a sensor array becomes vital. Therefore, a large-scale device integrated with 3 × 3 arrays was fabricated to monitor the spatial distributions of temperature and pressure by voltage and resistance signal mappings at different points. As shown in Fig. 5a, when a certain area of the sensor array is touched by an adult male's fingers, the touched pixels convert the temperature signal into a voltage signal based on the thermoelectric effect, and the distribution of the surface temperature can be recorded and revealed (Fig. 5b). Meanwhile, the spatial distribution of the resistance signals caused by the piezoresistance effect was recorded using histograms to obtain the spatial distribution of the external pressure stimuli (Fig. 5c). As the pressure of the touched area is different, the voltage values also vary because of the different contact thermal resistance, even though the finger is homoiothermic. In contrast, when two glass rods (near room temperature) were used to press the two units of the array, as observed in Fig. 5e, f, only a resistance signal was generated, and nearly no voltage signal was detected. Thus, it is clear that the sensor array can achieve real-time pressure/temperature mapping and a precise and rapid response owing to the excellent thermoelectric performance of the $Bi_2Te_3$ film and the novel design of the 3D-spiral structure.

## Discussion

The (000l)-textured $Bi_2Te_3$ is considered the most effective thermoelectric material near room temperature, producing both piezoresistive and thermoelectric effects, rendering it a suitable candidate

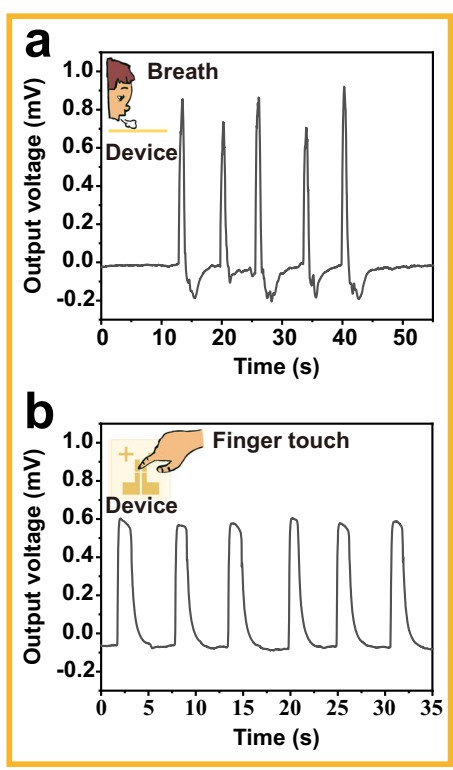
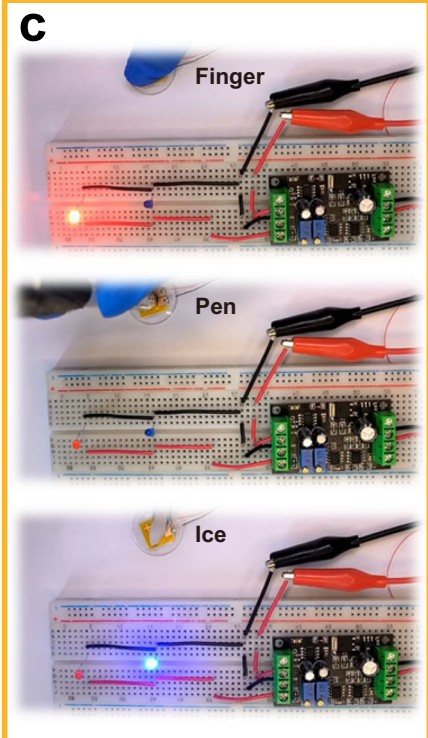
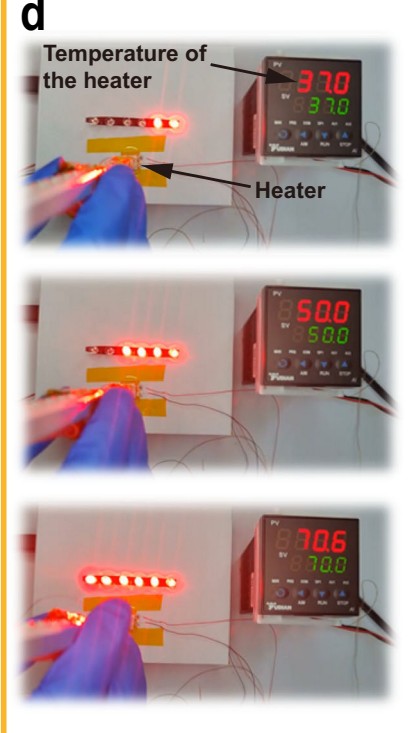

**Fig. 4 | Functional demonstration of the temperature sensor. a** Monitoring the human breath by testing the pulse voltage signal. **b** Thermoelectric responses to finger touch loading–unloading cycles. **c** Application of a sensor as a temperature switch, which can sense cold and warm to control an LED light. Upon touching the sensor with a finger, the red light activates. Conversely, there is no illumination when the sensor is touched with a pen. Additionally, placing an ice cube on the sensor triggers the blue light to illuminate. **d** Application of the sensor as a multistage temperature switch. Source data are provided as a Source Data file.

 

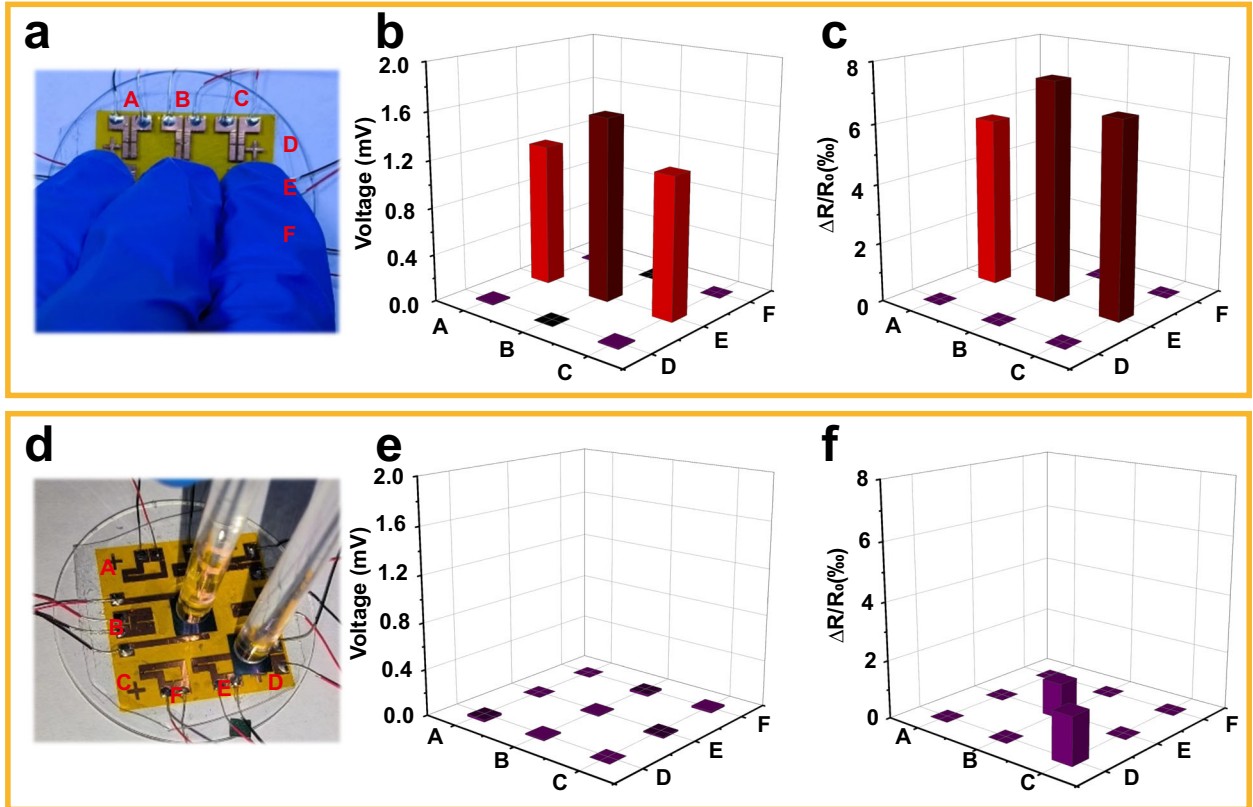

**Fig. 5 | Demonstration of pressure/temperature sensing function of the e-skin comprising 3 × 3 arrays pressed under different scenarios. a–c** Three gloved fingers with pressure and body temperature, and **d–f** two glass rods with pressure.

Spatial mapping of corresponding responses to pressure (**c**, **f**) and temperature (**b**, **e**) stimuli. The labels A to F represent the positional coordinates of the sensor array. Source data are provided as a Source Data file.

for pressure and temperature sensing applications. The piezoresistive effect may be owing to a change in the forbidden bandwidth caused by changes in the interlayer spacing changes under strain[33]. There is reason to believe that piezoresistive effects are present in textured $Bi_{1.5}Se_{0.5}Te_3$ and SnSe films, which will be investigated in future studies. The 3D-spiral structure is a universal structure appropriate for TE films to overcome the problem of thermal resistance matching, providing a new idea for further applications of TE films. By combining the dual-effect film and 3D-spiral structure, a high-performance pressure–temperature sensor can be fabricated and integrated into an array, demonstrating promising candidates for potential applications in skin-like intelligent devices. One limitation of this sensor is the temperature shift of the bottom surface when a relatively warmer object is in contact with the sensor[9]. Therefore, studies on calibrating the bottom temperature should be conducted to improve the sensor accuracy. This problem can be solved by integrating a micro Pt film resistance to monitor the bottom temperature of the sensor or to build appropriate mathematical models to correct the bottom temperature in specific cases.

We demonstrated the concept of constructing a TE temperature sensor that relies on a combination of a $Bi_2Te_3$ film and a special 3D-spiral design. This construction method is a simple but useful strategy for driving pressure and temperature sensors forward. The piezoresistive and TE effects of the $Bi_2Te_3$ films enable the detection of pressure and temperature stimuli. The high Seebeck coefficient imparts our sensor with a sensitivity as high as −426.4 μV K⁻¹, and the spiral structure reduces the response time to <1 s, which are promising results for future studies. Notably, the simple manufacturing process and excellent flexibility make it possible to map the temperature. The 3 × 3 sensor arrays are promising for use in wearable electronics, such as e-skin applications.

## Methods

### Materials and device preparation

*n*-Type $Bi_2Te_3$ and Au films were deposited on a 25-μm PI substrate by magnetron sputtering. Commercial $Bi_2Te_3$ (99.99%) and Te (99.99%) targets were used. The base pressure of the deposition chamber was <5 × 10⁻⁷ torr, 5-15 mT of Ar gas pressure for operation, and a deposition power of 40–60 W for $Bi_2Te_3$ and 30–50 W for Te were employed to optimize the performance of the film. Before deposition, the films were cut into special spiral patterns using a laboratory-built femtosecond laser. They were then cleaned with acetone, alcohol, and deionized water for 30 min in an ultrasonic bath. The deposition mask is designed as shown in Supplementary Figs. 8 and 9. After deposition, the Ag paste was dispensed using a lab-built semiautomatic dispensing machine (Supplementary Fig. 15 left). The Ag paste volume can be tuned by controlling dispenser time using a needle with a diameter of 65 μm. After dispensing, the Ag paste is cured at a 413 K hot plant for 30 min. Finally, the 3D-spiral structure and PDMS encapsulation were fabricated using laboratory-built device-integrated equipment, as shown in Supplementary Fig. 15 right, to complete the sensor.

### Measurement of the thermoelectric film and sensor

The microstructures of the samples were analyzed using scanning electron microscopy (SU-70, Hitachi) and X-ray diffraction (XRD; Ultimate IV, Rigaku). The in-plane Seebeck coefficient, $\alpha$, and electrical conductivity, $\sigma$, were measured by a Netzsch SBA-458 system under Ar + $H_2$ (5%) gas protection. The measurement errors for $\sigma$ and $\alpha$ were less than 5% and 3%, respectively. An HMS-5000 Hall system was employed to measure the Hall coefficient. The carrier concentration, $n$, at room temperature was determined on the assumption that the Hall coefficient, $R_H$, equals $1/n_e$, and the Hall mobility, $\mu_H$, was calculated using $\mu_H = \sigma/n_e$. A commercial pressure sensor (DZY-101, Bengbu Dayang

Sensing System Co., Ltd) was employed to measure the pressure and an alternating current resistance tester (AT526, Appellant Instruments Co., Ltd) was employed to record the sensor's resistance during the pressure-sensing test. During the temperature-sensing test, a DC power supply (ITech 6717 B) and PID control (AI-516, Xiamen Yudian Automation Technology Co., Ltd) were employed to control the temperature on one side of the sensor. The Seebeck voltage generated by the sensor was recorded using a Keithley 6500 digital multimeter. The pressure/temperature-sensing and bending tests were all performed using lab-built equipment (Supplementary Fig. 16).

### Reporting summary
Further information on research design is available in the Nature Portfolio Reporting Summary linked to this article.

## Data availability
The source data used in this study can be found in the Figshare database[52] and are available in the Source Data file. Source data are provided with this paper.

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

## Acknowledgements

This study is supported by the Ministry of Science and Technology of China (grant 2017YFA0700702, 2017YFA0700705), the National Natural Science Foundation of China (grants 52073290, 51927803), Liaoning Province science and technology plan project (2022-MS-011), Science Fund for Distinguished Young Scholars of Liaoning Province (2023JH6/100500004), Shenyang science and technology plan project (23-407-3-23).

## Author contributions

H.L.Y., Z.Y and K.P.T designed the research project and supervised the experiment. H.L.Y. and Z.Y. carried out experiments and analysed data. H.L.Y. wrote this manuscript. Z.Q.H, J.H., Y.J.R., Y.Z., Z.Y and K.P.T revised and edited this manuscript.

## Competing interests

The authors declare no competing interests.
