## [Peer Review File · Nature Communications]

Flexible temperature-pressure dual sensor based on 3D spiral thermoelectric Bi_2Te_3 filmsRESPONSE TO REVIEWERS' COMMENTS

Reviewer #1 (Remarks to the Author):

In this manuscript, the authors developed a multifunctional flexible sensor based on the piezoresistive and TE effects of Bi₂Te₃ thin films. It is interesting and the 3D spiral shaped structure is inspiring for the design of other sensors. However, due to some issues are not clear, this manuscript is recommended for publication after minor revisions.

(1) There is a lack of evidence that the (0001)-textured film are the key to achieving flexibility. A comparison with other structural films of the same thickness is required. Also, the thickness of the film may be an important factor, since even for the (0001)-textured films, larger thicknesses may lead to cracking, and the authors need to discuss these facts.

(2) Although the 3D spiral shaped structure is the key to the sensor, there are no images and size information about the 3D structure involved in the manuscript. Evidence for the 3D spiral shaped structure is required for further understanding. Also, the interconnection details between the 3 pairs also need to be provided.

(3) Line 163, "where d and r denote the thickness and radius of the PI film curvature," should be "where d and r denote the thickness and curvature radius of the PI film ,"

(4) Figure 1 b is not discussed in the manuscript.

(5) The response time of the sensor to temperature is provided in the manuscript. Similarly, the pressure response time also needs to be provided.

Reviewer #2 (Remarks to the Author):

This manuscript presents to fabricate 3D spiral thermoelectric Bi₂Te₃ films for flexible temperature-pressure dual sensors. This manuscript is well organized, but the following points need to be revised.

In the Introduction section, I recommend including references that show the piezoelectric effect of Bi₂Te₃ and the relationship between strain and Seebeck effect.

I recommend describing detailed photos of 3D spiral thermoelectric Bi₂Te₃ films in the manuscript.

I have also heated polyimide substrates; does the polyimide turn black and shrink at the heat treatment temperature of 613 K in your experiment? If not, are the treatment temperatures accurately measured?

Why did you use the combination of n-type Bi₂Te₃ and Au as the device structure? The performance should be enhanced by using a p-type film such as Sb₂Te₃ instead of Au.

In the XRD data in Supplementary Fig. 3, I am curious about the XRD pattern of the film at 613 K. If the film is c-axis orientated to the c-axis direction, peaks of (0015) and (0018) should be visible in addition to the (006) peak. In addition, the (003) peak should be visible at 8.6 degrees. I recommend that the authors show data from 8 degrees to show the peak of (003).

In the XRD data in Supplementary Fig. 3, increasing the heat treatment temperature causes the substrate to shrink, which may shift the XRD peaks towards the high-angle side. The authors should check the 006 and 015 peaks to see if a peak shift is due to substrate shrinkage.

Reviewer #3 (Remarks to the Author):

In this manuscript by Hailong, et.al., the authors report on a device based on thermoelectric Bi₂Te₃ that can detect both pressure and temperature. It was inferred that the main novelty here lies in the design (3D spiral device), which is supposed to overcome the tradeoff between low thermal impedance and low sensing performance (response time).

The objective and introduction in this work is clearly laid out, but I do not think the authors address the state-of-the-art problem. In the introduction, it was claimed that "However, the lateral structure design of most thin-film TE devices resulting in a long response time owing to the long and thin TE leg as a heat and current diffusing route", how long is long? And what is the response time in this work compared to literature? For a manuscript in high quality journal such as nature comm, one cannot make such unfounded, generalized claim like this without proper quantitative discussion. Likewise, "However, vertical thin-film devices face a significant challenge in creating a sufficient and stable temperature difference in the out-of-plane direction owing to its low thermal impedance." Again, no number is given, and no comparison between what is achieved in this work vs what was achieved. This makes the claim sound empty and makes it hard to evaluate the real contribution of this work. Therefore, I would strongly recommend rejection of the manuscript.

Detailed comments:

- I find it hard to believe that while the main novelty in this paper is 3D spiral structure, no picture of the actual sample was included anywhere in the manuscript/supporting information. Any micrograph (TEM, SEM, or even optical microscopy) showing the 3D spiral structure? It is very hard to envisage the benefit of such structure if the reader cannot even see it. Supplementary fig 1b does not look anything close to "3D spiral" to me. One should not use such fancy and misleading terms to describe their finding and to make it appear as if there is novelty.

- In theory, any degenerate semiconductor material (most thermoelectric compounds) should be piezoresistive, it should not be a unique properties of Bi₂Te₃. In the introduction, the authors claimed as if this is the first time such piezoresistive effect is discovered in Bi₂Te₃. A more careful approach should be taken in order not to overclaim their achievements.

- Many of the supplementary figures are not discussed in the main manuscript, only in the supporting information. I would suggest all the supplementary figures discussed in the main manuscript to come first, so as not to cause confusion.

- Literatures are outdated, many works cited are from the past decade, and only 5 out of 41 references are from the past 2 years. While this in itself is fine, this can be an indication that either this work is not novel, or the authors have lack of awareness of the state-of-art progress in the field. Either way, the authors seem to have very limited knowledge about thermoelectrics, and there seems to be no real science in this paper, mostly engineering and fanciful device/claim.

- Figure 1b and 1c appears contradictory. In Figure 1b, the change in resistance appears to be transient and time dependent, and no indication of strain level was presented in fig 1b. On the other hand, figure 1c is about strain vs change in resistance. My question is, since the change in resistance is time-dependent, how can fig 1c be determined? Worse still, figure 1D appears to contradict both 1c and 1b!!! how can $R/R_0 = 1$ for all radius of curvature (strain level)? Something sounds very wrong in this entire figure 1. This is a very elementary error and makes me wonder, how can I believe the data in this manuscript at all. Something is fishy here.

- Again, figure 3a is very unprofessional. It only shows the output (voltage vs time at a fixed dT), but what is the input signal? What is the dT vs time profile that is used to drive the output? Authors are not at all transparent in their presentation. In fig 3b, the sensitivity of 369.6 $\mu\text{V}/\text{K}$ does not seem to tally with the Seebeck coefficient (and number of pairs) of the device. Figure 3c seems to contradict the inset of fig 1d, are they from the same sample? And I need to understand, why is there a need to present similar experiment twice? Fig 3e is not valid, sensitivity can easily be increased by joining more thermoelectric junctions together, this is not intrinsic properties and is meaningless.

- In fig 5, the drawback of this kind of device is, resistance measurement need to be constantly on to measure any pressure change. In real life application, this does not seem to be practical.

- The home made dispensing device in Fig S9 appears to be very basic, how can the authors ensure minimal oxidation of the Bi₂Te₃ film? Usually the process is done in vacuum. More elaborate description of Fig S9 setup and details are needed.

Overall, I do not believe this article is of the level of Nature Communications. In my opinion, even with proper revision and additional experiments, this is at best Advanced Engineering Materials or Advanced Materials Technologies level of article, which falls short of the quality required for the very high standard for Nature Communications.

RESPONSE TO REVIEWERS' COMMENTS

Reviewer #1 (Remarks to the Author):

In this manuscript, the authors developed a multifunctional flexible sensor based on the piezoresistive and TE effects of Bi_2Te_3 thin films. It is interesting and the 3D spiral-shaped structure is inspiring for the design of other sensors. However, due to some issues are not clear, this manuscript is recommended for publication after minor revisions.

Response: We thank the reviewer for these supportive comments and appreciate the suggestions that help improve our manuscript.

(1) There is a lack of evidence that the (0001)-textured film are the key to achieving flexibility. A comparison with other structural films of the same thickness is required. Also, the thickness of the film may be an important factor, since even for the (0001)-textured films, larger thicknesses may lead to cracking, and the authors need to discuss these facts.

Response: We thank the reviewer for this valuable comment.

It is indeed that the (0001)-textured film is the key to achieving flexibility. Thus, to better explain this point, we have further explored the changes in relative resistance for the (0001) and (015) textured Bi_2Te_3 film of the same thickness with substrate thicknesses of 25 μm and 125 μm . As Figure R1a and b shows, the (015)-textured film with 125 μm thick substrate's resistance doubled after a single bend at the radius of 2.8 mm and increased to 75 times larger than the initial state after 1000 bend cycles. While the (0001)-textured film with 125 μm thick substrate's resistance increased only about 1.05 times after a single bend and 5 times after 1000 bend cycles. Therefore, the (0001) texture is good for flexibility.

Figure R1. (a) Relative electrical resistance as a function of bending radius for the film. (b) The resistance change of film under 1000 cycles bend test at a bend radius of 2.8 mm. Legend Description: The lower triangle and superior triangle represent the sample with (000) -texture and (015) -texture film deposited on a $25 \mu\text{m}$ thick substrate. The hexagons and pentagons represent the sample with (000) -texture and (015) -texture film deposited on a $125 \mu\text{m}$ thick substrate.

The flexibility is not only related to the thickness of the film but also to the thickness of the substrate. When a deposited film and a compliant substrate have the same Young's modulus, the strain in the top bending sample (ε_b) is as follows,

$$\varepsilon_b = \frac{d_{\text{sample}} + d_{\text{substrate}}}{2r_b} \quad (9)$$

Where d_{sample} and $d_{\text{substrate}}$ are the thickness of film and polyimide, respectively, and r_b is the bending radius. In this scenario, a thicker film under the same bending radius will result in greater strain. But in our case, Young's modulus of Bi_2Te_3 and polyimide is different. The ε_b should be given by¹⁻³:

$$\varepsilon_b = \frac{d_{\text{sample}} + d_{\text{substrate}}}{2r_b} \frac{1 + 2\eta + \chi\eta^2}{(1 + \eta)(1 + \chi\eta)} = \frac{d_{\text{substrate}}}{2r_b} \frac{1 + 2\eta + \chi\eta^2}{1 + \chi\eta} \quad (10)$$

Where $\eta = d_{\text{sample}}/d_{\text{substrate}}$ and $\chi = Y_{\text{sample}}/Y_{\text{substrate}}$, $Y_{\text{sample}} = 54.2 \text{ Gpa}$ and $Y_{\text{substrate}} = 2.5 \text{ Gpa}$ denote Young's modulus of the film and substrate^{4,5}.

As can be seen in Figure R2a, the coefficient $\frac{1+2\eta+\chi\eta^2}{1+\chi\eta}$ decreases with the $d_{\text{sample}}/d_{\text{substrate}}$ increase and then increase with the $d_{\text{sample}}/d_{\text{substrate}}$ increase. Until $d_{\text{sample}}/d_{\text{substrate}}$ reaches approximately 0.16, coefficient $\frac{1+2\eta+\chi\eta^2}{1+\chi\eta}$ reached the minimum. It means if the thickness of polyimide is $25 \mu\text{m}$, the ε_b will reach the minimum value when the $d_{\text{sample}} \approx 4 \mu\text{m}$, which can

effectively improve the flexibility. Because the thickness of the films we deposited is less than 4 μm , the flexibility will be better with the thickness increase theoretically.

The thickness of the substrate is another key point to achieving flexibility. Figure R2 (b-c) shows the stress nephogram of the film under different bend radii and substrate thicknesses. It is easy to know the strain where substrate = 125 μm is much larger than where $d_{\text{substrate}} = 25 \mu\text{m}$ under the same bend situation and d_{sample} . As can be seen, when the d_{sample} is close to 0, the strain in the film is about 3% at $d_{\text{substrate}} = 125 \mu\text{m}$ under a bend radius of about 2 mm, which is 5 times larger than strain ($\approx 0.6\%$) at $d_{\text{substrate}} = 25 \mu\text{m}$. Therefore, the smaller thickness of the substrate is a key parameter to obtain good flexible films. Those also can be proved in Figures R1a and b, where the resistance has barely changed in all samples with 25 μm thick substrate after the bend test.

Figure. R2 (a) the coefficient $\frac{1+2\eta+\chi\eta^2}{1+\chi\eta}$ as a function of $d_{\text{sample}}/d_{\text{substrate}}$. (b-c) Stress nephogram at different film thicknesses and bending radii under 25 μm (b) and 125 μm (c) thick substrate, respectively. The solid black lines represent the isostrain lines.

Corresponding changes made:

- (1) The following figure has been added to the revised supplementary file.

Supplementary Figure 5 | Estimation of strain during film bending and the flexibility test result of various Bi₂Te₃/polyimide films with the same d_{sample}.

(a) The coefficient $\frac{1+2\eta+\chi\eta^2}{1+\chi\eta}$ as a function of $d_{sample}/d_{substrate}$. (b-c) Stress nephogram at different film thicknesses and bending radii under 25 μm (b) and 125 μm (c) thick substrate, respectively. The solid black lines represent the isostrain lines. (d) Relative electrical resistance as a function of bending radius for the film. (e) The resistance change of film under 1000 cycles bend test at a bend radius of 2.8mm. Legend Description: The lower triangle and superior triangle represent the sample with (000l)-texture and (015)-texture film deposited on a 25 μm thick substrate. The hexagons and pentagons represent the sample with (000l)-texture and (015)-texture film deposited on a 125 μm thick substrate.

(2) The following descriptions have been added to the revised manuscript:

“Strain analysis and the flexible test are discussed in Supplementary Note 2, and the result is shown in Supplementary Fig. 5. The thickness of the film is an important factor in achieving flexibility. As can be seen in Supplementary Fig. 5a, the coefficient $\frac{1+2\eta+\chi\eta^2}{1+\chi\eta}$ decreases with the $d_{sample}/d_{substrate}$ increase and then increase with the $d_{sample}/d_{substrate}$ increase. Until $d_{sample}/d_{substrate}$ reaches approximately 0.16, coefficient $\frac{1+2\eta+\chi\eta^2}{1+\chi\eta}$ reached the minimum. It

means if the thickness of polyimide is 25 μm , the ε_b will reach the minimum value when the $d_{\text{sample}} \approx 4 \mu\text{m}$, which can effectively improve the flexibility. Cause the thickness of the films we deposited is less than 4 μm , the flexibility will be better with the thickness increase theoretically.” (Line 207, revised main text)

“Supplementary Fig. 5b and c shows the stress nephogram of the film under different bend radii and substrate thicknesses. It is easy to know the strain where $d_{\text{substrate}} = 125 \mu\text{m}$ is much larger than where $d_{\text{substrate}} = 25 \mu\text{m}$ under the same bend situation and d_{sample} . As can be seen, when the d_{sample} is close to 0, the strain in the film is about 3% at $d_{\text{substrate}}=125 \mu\text{m}$ under a bend radius of about 2 mm, which is 5 times larger than strain ($\approx 0.6\%$) at $d_{\text{substrate}} = 25 \mu\text{m}$.” (Line 217, revised main text)

“As Supplementary Fig. 5d and e shows, the (015)-textured film with 125 μm thick substrate’s resistance doubled after a single bend at the radius of 2.8 mm and increased to 75 times larger than the initial state after 1000 bend cycles. While the (000l)-textured film with 125 μm thick substrate’s resistance increased only about 5 times after 1000 bend cycles. So, (000l) texture is indeed conducive to improving the flexibility of Bi_2Te_3 films.” (Line 224, revised main text)

(3) The following descriptions have been added to the revised supplementary file.

“Supplementary Note 2 Strain analysis of the flexible test for Bi_2Te_3 /polyimide films

One of the primary strategies for enhancing the flexibility of thin films is to minimize internal strain during film deformation. Therefore, the analysis of internal strain during film bending is of significant importance. When a deposited film and a compliant substrate have the same Young’s modulus, the strain in the top bending sample (ε_b) is as follows,

$$\varepsilon_b = \frac{d_{\text{sample}} + d_{\text{substrate}}}{2r_b} \quad (9)$$

Where d_{sample} and $d_{\text{substrate}}$ are the thickness of film and polyimide, respectively, and r_b is the bending radius. In this scenario, a thicker film under the same bending radius will result in greater strain. But in our case, the modulus of Bi_2Te_3 and polyimide differs. The ε_b should be given by¹⁻³:

$$\varepsilon_b = \frac{d_{sample} + d_{substrate}}{2r_b} \frac{1 + 2\eta + \chi\eta^2}{(1 + \eta)(1 + \chi\eta)} = \frac{d_{substrate}}{2r_b} \frac{1 + 2\eta + \chi\eta^2}{1 + \chi\eta} \quad (10)$$

Where $\eta = d_{sample}/d_{substrate}$ and $\chi = Y_{sample}/Y_{substrate}$, $Y_{sample} = 54.2 \text{ Gpa}$ and $Y_{substrate} = 2.5 \text{ Gpa}$ denote Young's modulus of the film and substrate^{4,5}."

(2) Although the 3D spiral-shaped structure is the key to the sensor, there are no images and size information about the 3D structure involved in the manuscript. Evidence for the 3D spiral-shaped structure is required for further understanding. Also, the interconnection details between the 3 pairs need to be provided.

Response: We thank the reviewer for this valuable comment. To better clarify the 3D spiral shape, we further explored the structures by an optical photograph and X-ray tomography (XRT), which can reflect the internal structure information of the device (Figure R3)⁶. As the results show, a 3D spiral-shaped structure can be obtained, which proves the structure feature. As shown in Figure R3a, b, and c, the sensor contains 3-pair legs. Figure R3d shows the 3D structure of the spiral constructed by X-ray tomography (XRT), which shows an inverted tower-type spiral structure. Figure R3e further shows a cross-section of Figure 3d, where Bi_2Te_3 film, Ag paste, and Cu foil can be seen. It is worth noting that the Ag paste is tightly attached to the Cu foil, which indicates a good electrical connection.

Figure R3. (a, b) Optical photograph of the upside and downside of the 3-pair leg device. (c) Enlarge of the 3D-spiral structure. Scale bars, 2 mm. (d) A 3D spiral structure of one leg in a device constructed by XRT. Scale bars, 250 μm . (e) The corresponding cross-section of the spiral structure image of Supplementary Figure 6d by XRT. The red ellipse represents Bi_2Te_3 film, the bright line is Ag paste and the Cu foil is under Ag paste. Scale bars, 250 μm .

Furthermore, the interconnection details between the 3 pairs of legs are further clarified in Figure R4. Figure R4a and b shows the diagram of the sensor electrical interconnection. Figure R4b depicts a simplified schematic diagram of Figure R4a. The electrical interconnection of a sensor is constructed by three sets of π -type structures connected in series, with one set of π -type structures consisting of two copper foils (black lines) and a pair of interconnected spirals (red and blue spiral).

Figure R4. (a) Simplified schematic diagram of device electrical connections. (b) The interconnection details of the 3-pair device. The red spiral and blue spiral represent the Au film and the Bi_2Te_3 film, and the black line denotes the Cu foil used as an electrode.

Corresponding changes made:

- (1) The following figure has been added to the revised supplementary file.

Supplementary Figure 6 | The spiral structure of the sensor and its electrical connection. (a, b) Optical photograph of the upside and downside of the 3-pair leg device. (c) Enlarge of the 3D-spiral structure. Scale bars, 2 mm. (d) A 3D spiral structure of one leg in a device constructed by XRT. Scale bars, 250 μm . (e) The corresponding cross-section of the spiral structure image of Supplementary Figure 6d by XRT. The red ellipse represents Bi_2Te_3 film, the bright line is Ag paste and the Cu foil is under Ag paste. Scale bars, 250 μm . (f) Simplified schematic diagram of device electrical connections. (g) The interconnection details of the 3-pair device. The red spiral and blue spiral represent the Au film and the Bi_2Te_3 film, and the black line denotes the Cu foil used as an electrode.

(2) The following descriptions have been added to the manuscript:

“The structure features of a sensor and the electrical interconnection are shown in Supplementary Fig. 6. As can be seen in Supplementary Fig. 6a, b, and c, the sensor contains 3-pair legs. Figure R3d shows the 3D structure of the spiral constructed by X-ray tomography (XRT), which shows an inverted tower-type spiral structure. Supplementary Fig. 6e further shows the corresponding cross-section of the spiral structure image of Supplementary Figure 6d by XRT, where Bi_2Te_3 film, Ag paste, and Cu foil can be seen. It is worth noting that the Ag paste is tightly attached to the Cu foil, which indicates a good electrical connection. Supplementary Fig. 6f and g shows the diagram of the sensor electrical interconnection. Supplementary Fig. 6g depicts a simplified schematic diagram of Supplementary Fig. 6f. The electrical interconnection of a sensor constructed by three sets of π -type

structures connected in series, with one set of π -type structures consisting of two copper foils (black lines) and a pair of interconnected spirals (red and blue spiral).” (Line 242, revised main text)

(3) Line 163, “where d and r denote the thickness and radius of the PI film curvature,” should be “where d and r denote the thickness and curvature radius of the PI film,”

Response: We thank the reviewer for this valuable comment. The sentence “where d and r denote the thickness and radius of the PI film curvature” has been revised as “where d and r denote the thickness and curvature radius of the PI film”.

(4) Figure 1 b is not discussed in the manuscript.

Response: We thank the reviewer for this valuable comment. Figure. 1b has already been discussed in the manuscript as follows: “Fig. 1b shows the time-dependent evolution of the relative resistance ($\Delta R/R_0$) of the Bi_2Te_3 film in response to uniaxial strain, as the bending radius is gradually increased and then decreased. An obvious piezoresistive phenomenon is observed as the $\Delta R/R_0$ decreases under a positive bend curvature ($\rho > 0$), whereas $\Delta R/R_0$ increases under a negative curvature ($\rho < 0$).” However, to provide a clearer description of our experiment, we have made certain modifications to this statement.

Corresponding changes have been made:

(1) The following descriptions have been added to the revised manuscript:

“The curve displays numerous steps, each representing a bending radius, which is identified through camera image recognition. The $\Delta R/R_0$ presents a stepwise decrease with the increase of stepwise bending radius under positive curvature, while presents a stepwise increase with the decrease of stepwise bending radius under positive curvature. The bend radii are 6.3 mm, 4.7 mm, 4 mm, 3.4 mm, and 2.9 mm, respectively. In the case of negative curvature bending, the resistance change exhibits the opposite behavior. The bend radii are 5.4 mm, 4.9 mm, 4 mm, 3.4 mm, and 3.1 mm, respectively. So, an obvious piezoresistive phenomenon is observed as the $\Delta R/R_0$ decreases under

a positive bend curvature ($\rho > 0$), whereas $\Delta R/R_0$ increases under a negative curvature ($\rho < 0$)."

(Line 177, revised main text)

(2) Figure. 1 has been revised as follows:

Fig. 1 Electric and flexible properties of Bi₂Te₃/PI films deposited at 613 K. (a)

Temperature-dependent Seebeck coefficient, electrical conductivity, and power factor of the film. (b)

Real-time piezoresistance response curve of the film. The numbers indicated by the arrows represent the

bending radius. (c) Relative resistance changes of the film as a function of strain. The red line is the fitted

curve, whose slope denotes the gauge factor of -9.1922. (d) Relative electrical resistance as a function of

bending radius for the film. R_a and R_0 denote the resistance of the film after bending deformation and the

original flat state, respectively. Inset: results of the cyclic bending test under $r = 2.8$ mm for the film.

(5) The response time of the sensor to temperature is provided in the manuscript. Similarly, the pressure response time also needs to be provided.

Response: We thank the reviewer for this valuable comment. We have tested the pressure response time of the sensor. Figure R5 shows the resistance change under 3 times 5.3 kPa load and unload process. The curve shows a stable response for the pressure sense and exhibits an instant response time of ~ 0.3 s and a recovery time of ~ 1.1 s.

Figure R5. The time-resolved response of the sensor to pressure stimuli with the red and blue zones corresponds to the response time and recovery time. The 3 types of color of the curve represent 3 times the test for the sensor.

Corresponding changes made:

(1) The following descriptions have been added to the revised manuscript:

“Further, we have tested the pressure response time of the sensor. Supplementary Fig. 11 shows the resistance change under 3 times 5.3 kPa load and unload process. The curve shows a stable response for the pressure sense and exhibits an instant response time of ~ 0.3 s and a recovery time of ~ 1.1 s.”
(Line 364, revised main text)

(2) Figure R5 has been added in the revised Supplementary Figure. 11 as follows:

Supplementary Figure 11 | The time-resolved response of the sensor to pressure stimuli with the red and blue zones corresponding to the response time and recovery time. The 3 types of color of the curve represent 3 times the test for the sensor.

Reviewer #2 (Remarks to the Author):

This manuscript presents to fabrication of 3D spiral thermoelectric Bi_2Te_3 films for flexible temperature-pressure dual sensors. This manuscript is well organized, but the following points need to be revised.

Response: We thank the reviewer for these supportive comments and appreciate the suggestions that help improve our manuscript.

1. In the Introduction section, I recommend including references that show the piezoelectric effect of Bi_2Te_3 and the relationship between strain and the Seebeck effect.

Response: We thank the reviewer for this valuable comment. We reported the piezoresistive effect of Bi_2Te_3 rather than the piezoelectric effect.

Currently, most of the research on Bi_2Te_3 focuses on enhancing its thermoelectric performance, with very little research dedicated to its piezoresistive effects. This may be attributed to the current research emphasis on new energy materials. Additionally, due to the brittleness of Bi_2Te_3 , measuring the piezoresistive effects of bulk materials can be challenging. The article⁷ published in 2023 has mentioned the piezoresistive effect of Bi_2Te_3 , and its gauge factor (GF) was reported to be -1.12, which is lower than our result of -9.19. We have already cited this study in our manuscript.

Furthermore, the relationship between strain and the Seebeck effect in Bi_2Te_3 has been extensively studied. Weiliang⁸ and Hajji⁹ utilized first principles to investigate the impact of biaxial mechanical strains (compression and tensile) on the thermoelectric performance of Bi_2Te_3 . Both studies found that the Seebeck coefficient increases under compressive strain and decreases under tensile strain.

Corresponding changes made:

The following descriptions have been added to the manuscript:

“However, the Seebeck coefficient of Bi_2Te_3 will be changed under the strain, which is studied by Weiliang⁸ and Hajji⁹. They used first principles to study the effect of thermoelectric performance under the biaxial mechanical strains (pressure and tensile). They both found that the Seebeck coefficient increases under compressive strain and decreases under tensile strain. These theoretical results remind us of the need to pay attention to the effect of strain on the Seebeck coefficient and to correct it as much as possible when using Bi_2Te_3 for temperature and pressure sensing.” (Line 120, revised main text)

2. I recommend describing detailed photos of 3D spiral thermoelectric Bi_2Te_3 films in the manuscript.

Response: We thank the reviewer for this valuable comment. To better clarify the 3D spiral shape, we further explored the structures by an optical photograph and X-ray tomography (XRT), which can reflect the internal structure information of the device (Figure R3) ⁶. As the results show, a 3D spiral-shaped structure can be obtained, which proves the structure feature. As shown in Figure R3a, b, and c, the sensor contains 3-pair legs. Figure R3d shows the 3D structure of the spiral constructed by X-ray tomography (XRT), which shows an inverted tower-type spiral structure. Figure R3e further shows a cross-section of Figure 3d, where Bi_2Te_3 film, Ag paste, and Cu foil can be seen. It is worth noting that the Ag paste is tightly attached to the Cu foil, which indicates a good electrical connection.

Figure R3. (a, b) Optical photograph of the upside and downside of the 3-pair leg device. (c) Enlarge of the 3D-spiral structure. Scale bars, 2 mm. (d) A 3D spiral structure of one leg in a device constructed by XRT. Scale bars, 250 μm . (e) The corresponding cross-section of the spiral structure image of Supplementary Figure 6d by XRT. The red ellipse represents Bi_2Te_3 film, the bright line is Ag paste and the Cu foil is under Ag paste. Scale bars, 250 μm .

Furthermore, the interconnection details between the 3 pairs of legs are further clarified in Figure R4. Figure R4a and g shows the diagram of the sensor electrical interconnection. Figure R4b depicts a simplified schematic diagram of Figure R4a. The electrical interconnection of a sensor is constructed by three sets of π -type structures connected in series, with one set of π -type structures consisting of two copper foils (black lines) and a pair of interconnected spirals (red and blue spiral).

Figure R4. (a) Simplified schematic diagram of device electrical connections. (b) The interconnection details of the 3-pair device. The red spiral and blue spiral represent the Au film and the Bi_2Te_3 film, and the black line denotes the Cu foil used as an electrode.

Corresponding changes made:

- (1) The following figure has been added to the supplementary file.

Supplementary Figure 6 | The spiral structure of the sensor and its electrical connection. (a, b) Optical photograph of the upside and downside of the 3-pair leg device. Scale bars, 2 mm. (c) Enlarge of the 3D-spiral structure. (d) A 3D spiral structure of one leg in a device constructed by XRT. Scale bars, 250 μm . (e) The corresponding cross-section of the spiral structure image of Supplementary Figure 6d by XRT. The red ellipse represents Bi_2Te_3 film, the bright line is Ag paste and the Cu foil is under Ag paste. Scale bars, 250 μm . (f) Simplified schematic diagram of device electrical connections. (g) The interconnection details of the 3-pair device. The red spiral and blue spiral represent the Au film and the Bi_2Te_3 film, and the black line denotes the Cu foil used as an electrode.

(2) The following descriptions have been added to the manuscript:

“The structure features of a sensor and the electrical interconnection are shown in Supplementary Fig. 6. As can be seen in Supplementary Fig. 6a, b, and c, the sensor contains 3-pair legs. Supplementary Fig. 6d shows the 3D structure of the spiral constructed by X-ray tomography (XRT), which shows an inverted tower-type spiral structure. Supplementary Fig. 6e further shows the corresponding cross-section of the spiral structure image of Supplementary Figure 6d by XRT, where Bi_2Te_3 film, Ag paste, and Cu foil can be seen. It is worth noting that the Ag paste is tightly attached to the Cu foil, which indicates a good electrical connection. Supplementary Fig. 6f and g shows the diagram of the sensor electrical interconnection. Supplementary Fig. 6g depicts a simplified schematic diagram of Supplementary Fig. 6f. The electrical interconnection of a sensor

constructed by three sets of π -type structures connected in series, with one set of π -type structures consisting of two copper foils (black lines) and a pair of interconnected spirals (red and blue spiral).”

(Line 242, revised main text)

3. I have also heated polyimide substrates; does the polyimide turn black and shrink at the heat treatment temperature of 613 K in your experiment? If not, are the treatment temperatures accurately measured?

Response: We thank the reviewer for this valuable comment. We used the Kapton HN type polyimide made by DuPont company, which could resist high temperatures of $< 773 \text{ K}^{10}$. So we didn't observe polyimide turn black and shrink after deposition. In our experiment, The deposition temperature was measured by a K-type thermocouple near the sample. the film was heated to 613 K in 15 K min^{-1} and then held for two hours in the magnetron sputtering chamber.

4. Why did you use the combination of n-type Bi_2Te_3 and Au as the device structure? The performance should be enhanced by using a p-type film such as Sb_2Te_3 instead of Au.

Response: We thank the reviewer for this valuable comment. First, our work mainly focuses on designing and realizing 3D-spiral structure devices, which achieve thermal impedance matching in the vertical direction. A low-cost, straightforward, and effective method is to use one kind of thermoelectric material to achieve it. Meanwhile, the dual-parameter pressure and temperature sense can be achieved using Bi_2Te_3 . Furthermore, the sensitivity of the sensor has reached a high level compared to other studies. Indeed, using the Sb_2Te_3 would enhance temperature sensing performance, but the piezoresistive effect of Sb_2Te_3 also needs to be considered. Our future research endeavors may involve the incorporation of p-type thermoelectric materials and the investigation of their piezoresistive effects.

5. In the XRD data in Supplementary Fig. 3, I am curious about the XRD pattern of the film at 613

K. If the film is c-axis orientated to the c-axis direction, peaks of (0015) and (0018) should be visible in addition to the (006) peak. In addition, the (003) peak should be visible at 8.6 degrees. I recommend that the authors show data from 8 degrees to show the peak of (003).

Response: We thank the reviewer for this valuable comment. New samples were prepared and characterized by Rigaku Ultimate IV powder X-ray, with Cu k_{α} radiation ($\lambda=1.5418\text{\AA}$) in a 2θ range of $5-90^{\circ}$. The result is shown in Figure R7. In the new spectrum, the (003), (006), (0015), and (0018) peak of the film deposited at 613 K is visible.

Figure R7. (a) The XRD spectrum of the films deposited at 298 K, 473 K, and 613 K.

6. In the XRD data in Supplementary Fig. 3, increasing the heat treatment temperature causes the substrate to shrink, which may shift the XRD peaks towards the high-angle side. The authors should check the 006 and 015 peaks to see if a peak shift is due to substrate shrinkage.

Response: We thank the reviewer for this valuable comment. Our results show that the (006) and (015) peaks shift towards the low-angle side, which means that the shrinking of the substrate is not the main reason. It is worth noting that the deposition temperature is not the heat treatment temperature, during the growth of Bi₂Te₃ films, there will be thermal stress due to the difference between Bi₂Te₃ and polyimide linear thermal expansion coefficients (LTECs). The LTECs of Bi₂Te₃¹¹ and polyimide are $17.11 \times 10^{-6} \text{ K}^{-1}$ and $20 \times 10^{-6} \text{ K}^{-1}$, so the compressive stresses inside the film will be created. The compressive stresses applied in the plane of the substrate increases the out-of-plane lattice constant, so our (006) and (015) peak move towards the lower angle with the

temperature increase.

Figure R8. (a) Enlarge of the XRD spectrum near (006) peak. (b) Enlarge of the XRD spectrum near (015) peak.

Corresponding changes made:

(1) The following descriptions have been added to the manuscript:

“In this study, a high deposition temperature of 613 K is implemented and a high crystal quality Bi_2Te_3 film with a $(000l)$ texture was achieved to improve the electrical conductivity, which can be identified in the scanning electron microscopy image in Supplementary Fig. 1c, the Electron Back-Scattered Diffraction (EBSD) image in Supplementary Fig. 2 and the X-ray diffraction (XRD) pattern in Supplementary Fig. 3d and Supplementary Fig. 4. Due to the different linear thermal expansion coefficients (LTECs) between Bi_2Te_3 ($17.11 \times 10^{-6} \text{ K}^{-1}$)¹¹ and polyimide ($20 \times 10^{-6} \text{ K}^{-1}$), the in-plane compressive stresses created inside the film. So, the (006) and (015) peak shift towards the lower angle with the temperature increase in the spectrum.” (Line 150, revised main text)

(2) The following figure has been added to Supplementary Fig. 4

Supplementary Figure 4 | (a) The XRD spectrum of the films deposited at 298 K, 473 K, and 613 K. (b) Enlarge of the spectrum near (006) peak. (c) Enlarge of the spectrum near (015) peak.

(1) The Supplementary Figure 3d has been deleted.

Reviewer #3 (Remarks to the Author):

In this manuscript by Hailong, et al., the authors report on a device based on thermoelectric Bi₂Te₃ that can detect both pressure and temperature. It was inferred that the main novelty here lies in the design (3D spiral device), which is supposed to overcome the tradeoff between low thermal impedance and low sensing performance (response time).

The objective and introduction in this work are laid out, but I do not think the authors address the state-of-the-art problem. In the introduction, it was claimed that “However, the lateral structure design of most thin-film TE devices resulting in a long response time owing to the long and thin TE leg as a heat and current diffusing route”, how long is long? And what is the response time in this work compared to literature? For a manuscript in a high-quality journal such as nature comm, one cannot make such an unfounded, generalized claim this without proper quantitative discussion. Likewise, “However, vertical thin-film devices face a significant challenge in creating a sufficient and stable temperature difference in the out-of-plane direction owing to its low thermal impedance.” Again, no number is given, and no comparison between what is achieved in this work vs. what was achieved. This makes the claim sound empty and makes it hard to evaluate the real contribution of this work. Therefore, I would strongly recommend the rejection of the manuscript.

Response: After having carefully read the comments and concerns. We believe that there must be some misunderstandings as we have not sufficiently elaborated on some issues, so please allow us to make some explanations regarding the following concerns.

First, the issue of long response times in thin-film, in-plane thermoelectric devices due to the thin and elongated nature of the thermoelectric legs is commonly observed. As reported by Huixu¹² and Ya¹³, the length of the thermoelectric legs in the device is 10 mm and 6 mm and the corresponding response time is 34 s and 17 s respectively. Typically, for planar thermoelectric devices (except for microdevices), the length of the thermoelectric leg is in the millimeter range. While the thickness of the film is typically only in the range of a few hundred nanometers to several tens of micrometers. Therefore, adopting a vertical structure can significantly reduce the response

time. Yingming¹⁴ used a SnSe film with 366 nm thick to realize an ultrafast infrared detection with a response time of 11.3 μ s. Second, researchers generally concur that low thermal impedance in the vertical direction in thin films makes it difficult to establish a temperature difference. We have cited two references^{15,16} in our manuscripts, both of which focused on establishing 3D structures to overcome thermal impedance matching issues. Third, the statement “Another drawback for the parallel TE device is the long response time due to the long and thin TE leg as heat and current diffusing route” published¹⁷ in “Advance Energy Materials” and “Thin films have a minimal thermal impedance in this direction due to their small thicknesses such that the temperature drops across this direction is, in practical terms of thermoelectric energy harvesting, negligible” published¹⁵ in “Science Advance”, align with our viewpoint. However, for clearer and better elucidation of our standpoint, we have made the following modifications.

Corresponding changes made:

(1) The following descriptions have been deleted:

“The fabrication of TE devices with vertical structures can effectively overcome these limitations owing to the shorter heat-diffusion route.” (Line 89, revised main text)

(2) The following descriptions have been added to the manuscript:

“As reported by Huixu¹⁷ and Ya¹⁸, where the length of the device is 10 mm and 6 mm, their response time is 34 seconds and 17 seconds. Typically, for planar thermoelectric devices (except for microdevices), the length of the thermoelectric leg is in the millimeter range. The thickness of the film is typically only in the range of a few hundred nanometers to several tens of micrometers. Therefore, adopting a vertical structure can significantly reduce the response time. Yingming¹⁹ used a SnSe film 366 nm thick to realize an ultrafast infrared detect with a response time of 11.3 μ s.” (Line 89, revised main text)

Detailed comments:

1. I find it hard to believe that while the main novelty in this paper is the 3D spiral structure, no picture of the actual sample was included anywhere in the manuscript/supporting information. Any micrograph (TEM, SEM, or even optical microscopy) showing the 3D spiral structure? It is very hard to envisage the benefit of such a structure if the reader cannot even see it. Supplementary fig 1b does not look anything close to a “3D spiral” to me. One should not use such fancy and misleading terms to describe their finding and to make it appear as if there is a novelty.

Response: We thank the reviewer for this valuable comment. To better clarify the 3D spiral shape, we further explored the structures by an optical photograph and X-ray tomography (XRT), which can reflect the internal structure information of the device (Figure R3)⁶. As the results show, a 3D spiral-shaped structure can be obtained, which proves the structure feature. As shown in Figure R3a, b, and c, the sensor contains 3-pair legs. Figure R3d shows the 3D structure of the spiral constructed by X-ray tomography (XRT), which shows an inverted tower-type spiral structure. Figure R3e further shows a cross-section of Figure 3d, where Bi_2Te_3 film, Ag paste, and Cu foil can be seen. It is worth noting that the Ag paste is tightly attached to the Cu foil, which indicates a good electrical connection.

Figure R3. (a, b) Optical photograph of the upside and downside of the 3-pair leg device. (c) Enlarge of the 3D-spiral structure. Scale bars, 2 mm. (d) A 3D spiral structure of one leg in a device constructed by XRT. Scale bars, 250 μm . (e) The corresponding cross-section of the spiral structure image of Supplementary Figure 6d by XRT. The red ellipse represents Bi_2Te_3 film, the bright line is Ag paste and the Cu foil is under Ag paste. Scale bars, 250 μm .

Furthermore, the interconnection details between the 3 pairs of legs are further clarified in Figure R4. Figure R4a and g shows the diagram of the sensor electrical interconnection. Figure R4b depicts a simplified schematic diagram of Figure R4a. The electrical interconnection of a sensor is constructed by three sets of π -type structures connected in series, with one set of π -type structures consisting of two copper foils (black lines) and a pair of interconnected spirals (red and blue spiral).

Figure R4. (a) Simplified schematic diagram of device electrical connections. (b) The interconnection details of the 3-pair device. The red spiral and blue spiral represent the Au film and the Bi_2Te_3 film, and the black line denotes the Cu foil used as an electrode.

Corresponding changes made:

- (1) The following figure has been added to the supplementary file.

Supplementary Figure 6 | The spiral structure of the sensor and its electrical connection. (a, b) Optical photograph of the upside and downside of the 3-pair leg device. Scale bars, 2 mm. (c) Enlarge of the 3D-spiral structure. (d) A 3D spiral structure of one leg in a device constructed by XRT. Scale bars, 250 μm . (e) The corresponding cross-section of the spiral structure image of Supplementary Figure 6d by XRT. The red ellipse represents Bi_2Te_3 film, the bright line is Ag paste and the Cu foil is under Ag paste. Scale bars, 250 μm . (f) Simplified schematic diagram of device electrical connections. (g) The interconnection details of the 3-pair device. The red spiral and blue spiral represent the Au film and the Bi_2Te_3 film, and the black line denotes the Cu foil used as an electrode.

(2) The following descriptions have been added to the manuscript:

“The structure features of a sensor and the electrical interconnection are shown in Supplementary Fig. 6. As can be seen in Supplementary Fig. 6a, b, and c, the sensor contains 3-pair legs. Figure R3d shows the 3D structure of the spiral constructed by X-ray tomography (XRT), which shows an inverted tower-type spiral structure. Supplementary Fig. 6e further shows the corresponding cross-section of the spiral structure image of Supplementary Figure 6d by XRT, where Bi_2Te_3 film, Ag paste, and Cu foil can be seen. It is worth noting that the Ag paste is tightly attached to the Cu foil, which indicates a good electrical connection. Supplementary Fig. 6f and g shows the diagram of the sensor electrical interconnection. Supplementary Fig. 6g depicts a simplified schematic diagram of

Supplementary Fig. 6f. The electrical interconnection of a sensor constructed by three sets of π -type structures connected in series, with one set of π -type structures consisting of two copper foils (black lines) and a pair of interconnected spirals (red and blue spiral).” (Line 242, revised main text)

2. In theory, any degenerate semiconductor material (most thermoelectric compounds) should be piezoresistive, it should not be a unique properties of Bi_2Te_3 . In the introduction, the authors claimed as if this is the first time such a piezoresistive effect has been discovered in Bi_2Te_3 . A more careful approach should be taken in order not to overclaim their achievements.

Response: We thank the reviewer for this valuable comment.

Currently, most of the research on Bi_2Te_3 focuses on enhancing its thermoelectric performance, with very little research dedicated to its piezoresistive effects. This may be attributed to the current research emphasis on new energy materials. Additionally, due to the brittleness of Bi_2Te_3 , measuring the piezoresistive effects of bulk materials can be challenging. The article⁷ published in 2023 has mentioned the piezoresistive effect of Bi_2Te_3 , and its gauge factor (GF) was reported to be -1.12, which is lower than our result of -9.19. We have already cited this study in our manuscript. To avoid any misunderstandings, we have made some revisions to our statements.

Corresponding changes made:

(1) The following descriptions have been revised:

The “Bismuth telluride (Bi_2Te_3), exhibiting the best TE performance near room temperature, also exhibits good piezoresistive effects we discovered” has been revised to “Bismuth telluride (Bi_2Te_3), exhibiting the best TE performance near room temperature, also exhibits good piezoresistive effects”. (Line 117, revised main text)

The “In addition, owing to the piezoresistive effect we discovered in the Bi_2Te_3 film” has been revised to “In addition, owing to the piezoresistive effect in the Bi_2Te_3 film”. (Line36, revised main text)

3. Many of the supplementary figures are not discussed in the main manuscript, only in the supporting information. I would suggest all the supplementary figures discussed in the main manuscript come first, so as not to confuse.

Response: We thank the reviewer for this valuable comment.

(1) In supplementary figures 1, 2, 3, and 4, the microstructure and thermoelectric performance of the films at different deposition temperatures are primarily showcased, which are discussed in supplementary note 1. We have revised the manuscripts and mentioned supplementary figures 1-4 in the manuscripts.

Corresponding changes made:

“The TE performance of the Bi_2Te_3 film was optimized and the details are provided in Supplementary Note 1 and Supplementary Fig.1-4.” (Line 147, revised main text)

“In this study, a high deposition temperature of 613 K is implemented and a high crystal quality Bi_2Te_3 film with a $(000l)$ texture was achieved to improve the electrical conductivity, which can be identified in the scanning electron microscopy image in Supplementary Fig. 1c, the Electron Back-Scattered Diffraction (EBSD) image in Supplementary Fig. 2 and the X-ray diffraction (XRD) pattern in and Supplementary Fig. 4.” (Line 150, revised main text)

(2) Supplementary Figure 7 shows the detailed manufacturing step of the sensor. They have already been described in Supplementary Note 2. We have also mentioned a simple simplified fabrication process in the manuscripts as follows:

Corresponding changes made:

“PDMS can be divided into three layers: the top encapsulation, intermediate filling, and bottom supporting layers (Supplementary Note 3 and Supplementary Fig. 7). (Line 240, revised main text)

(3) Supplementary Figures 8 and 9 have been mentioned in the revised manuscripts.

Corresponding changes made:

“The deposition mask is designed as shown in Supplementary Fig. 8 and 9.” (line 461, revised main text)

(4) Supplementary Figure 10 has been mentioned in the revised manuscripts.

Corresponding changes made:

“The temperature-sensing resolution was further evaluated, using the measurement device shown in Supplementary Fig. 10a. A sensor is attached to a surface of Thermo Electric Cooler (TEC) covered with embedded thermistor silicone, and the up surface of the sensor is in contact with a custom-made copper block, which temperature is controlled by a PID controller. The pressure sensor is installed under the water-cooling system which is to cool the TEC hot surface when it works. By keeping the copper block temperature constant at 303 K and adjusting the TEC current, a temperature difference is created between the upper and lower surfaces of the device. Supplementary Fig. 10b shows the sensitivity of the sensor under different pressures. The detail is discussed in Supplementary Note 4. As shown in Supplementary Fig. 10c, a temperature gradient of 0.1 K can be observed with steps shown in the V-t curve, indicating that the resolution can reach 0.1 K at room temperature.” (Line 333, revised main text)

“Supplementary Fig. 10b shows how the contact thermal resistance affects the measurement sensitivity. Under the pressure of 0g, the measurement sensitivity (S_{ob_su}) is only 13.19 $\mu\text{V K}^{-1}$, which is not the intrinsic parameter of the sensor. With the increase in pressure, the measurement sensitivity, S_{ob_su} , increases. When the pressure reaches 350 g, the S_{ob_su} reaches 131.4 $\mu\text{V K}^{-1}$. It is worth noting that we employed T_{ob} and T_{su} (temperature of the copper contacted with the sensor's upper surface and temperature of the thermally conductive silicone contacted with the sensor's lower surface) rather than T_s and T_o (temperatures of the upper and lower surfaces of the sensor) to calculate the ΔT . Due to the presence of unavoidable R_{ob} and R_{su} , as mentioned in equation (8), the measurement

sensitivity, S_{ob-sit} , is less than the actual sensitivity, S .” (line 329, Supplementary Note 4)

(5) Supplementary Figure 13c has been mentioned in revised manuscripts.

Corresponding changes made:

“Supplementary Figure 13c shows the sensor’s voltage response to temperature difference stimuli from 9.1K to 85.0 K.” (Line 315, revised main text)

(6) Supplementary Figure 18 has been mentioned in the revised manuscript.

Corresponding changes made:

“We build a simple one-dimensional heat conduction model as shown in Supplementary Note 4 and Supplementary Fig. 18, so the temperature-sensing sensitivity of the sensor is defined according to the formula:

$$S = A\alpha = \left(1 - \frac{R_{s1} + R_{01}}{R_{th} + R_{s1} + R_{01}}\right)\alpha \quad (1) \text{” (Line 272, revised main$$

text)

4. Literature is outdated, many works cited are from the past decade, and only 5 out of 41 references are from the past 2 years. While this in itself is fine, this can be an indication that either this work is not novel, or the authors have a lack of awareness of the state-of-art progress in the field. Either way, the authors seem to have very limited knowledge about thermoelectrics, and there seems to be no real science in this paper, mostly engineering and fanciful devices/claims.

Response: We thank the reviewer for this valuable comment. We have added and modified some references.

5. Figures 1b and 1c appear contradictory. In Figure 1b, the change in resistance appears to be transient and time-dependent, and no indication of strain level was presented in Figure 1b. On the

other hand, Figure 1c is about strain vs. change in resistance. My question is, since the change in resistance is time-dependent, how can Fig 1c be determined? Worse still, Figure 1d appears to contradict both 1c and 1b!!! how can $R/R_0 = 1$ for all radius of curvature (strain level)? Something sounds very wrong in this entire Figure 1. This is a very elementary error and makes me wonder, how can I believe the data in this manuscript at all. Something is fishy here.

Response: We thank the reviewer for this valuable comment. The change in resistance is not time-dependent but related to the strain in the film. In our experiment, the strain was applied to the Bi_2Te_3 film by controlling its bending (up or down) through a motorized translation stage, and the bending radius was finely controlled by tuning the moving steps using the bend test system shown in Supplementary Fig. 16. In Figure 1b, the curve displays numerous steps, each of them representing a bending radius, which is identified through camera image recognition. The $\Delta R/R_0$ presents a stepwise increase with the increasing bending radius and a decrease with the decreasing bending radius under positive curvature. The bend radii are 6.3 mm, 4.7 mm, 4 mm, 3.4 mm, and 2.9 mm, respectively. In the case of negative curvature bending, the resistance change exhibits the opposite behavior. And the bend radii are 5.4 mm, 4.9 mm, 4 mm, 3.4 mm, and 3.1 mm, respectively. (The bend radius has been indicated in revised Figure 1b.) The strain in the film can be estimated by $\varepsilon = d/2r$, where d and r denote the thickness and curvature radius of the PI film. So, the relationship between the strain and the resistance change can be plotted in Figure 1c.

In Figure 1d, R/R_0 shows the resistance after bending rather than the resistance under bending. Therefore, the results in Figure 1c and Figure 1d are not contradictory. We have revised the R/R_0 to R_a/R_0 .

Corresponding changes made:

(1) The following descriptions have been added to the revised:

“Strain was applied to the Bi_2Te_3 film by controlling its bending (up or down) through a motorized translation stage, and the bending radius was finely controlled by tuning the moving steps using the bend test system shown in Supplementary Fig. 16.” (Line 171, revised main text)

“The curve displays numerous steps, each representing a bending radius, which is identified through

camera image recognition. The $\Delta R/R_0$ presents a stepwise increase with the increasing of bending radius under positive curvature while presenting a stepwise decrease with the decrease of stepwise bending radius under positive curvature. The bend radii are 6.3 mm, 4.7 mm, 4 mm, 3.4 mm, and 2.9 mm, respectively. In the case of negative curvature bending, the resistance change exhibits the opposite behavior. And the bend radii are 5.4 mm, 4.9 mm, 4 mm, 3.4 mm, and 3.1 mm, respectively.” (Line 177, revised main text)

“Referring to the original resistance, R_0 , the normalized resistance (R_a/R_0) after a single bend is plotted in Fig. 1d as a function of the bend radius.” (Line 202, revised main text)

(2) Figure 1b has been revised, follows shows the revised Figure 1.

Fig. 1 Electric and flexible properties of Bi₂Te₃/PI films deposited at 613K. (a) Temperature-dependent Seebeck coefficient, electrical conductivity, and power factor of the film. (b) Real-time piezoresistance response curve of the film. (c) Relative resistance changes of the film as a function of strain. The red line is the fitted curve, whose slope denotes the gauge factor of -9.1922. (d)

Relative electrical resistance as a function of bending radius for the film. R_a and R_0 denote the resistance of the film after bending deformation and the original flat state, respectively. Inset: results of the cyclic bending test under $r = 2.8$ mm for the film.

6. Again, figure 3a is very unprofessional. It only shows the output (voltage vs. time at a fixed ΔT), but what is the input signal? What is the ΔT vs. time profile that is used to drive the output? The authors are not at all transparent in their presentation. In Fig 3b, the sensitivity of $369.6 \mu\text{V/K}$ does not seem to tally with the Seebeck coefficient (and number of pairs) of the device. Figure 3c seems to contradict the inset of Fig 1d, are they from the same sample? I need to understand, why is there a need to present a similar experiment twice? Fig 3e is not valid, sensitivity can easily be increased by joining more thermoelectric junctions together, this is not intrinsic properties and is meaningless.

Response: We thank the reviewer for this valuable comment. Fig. 3a shows the sensing performance with five on/off cycles for each ΔT , which is controlled by a hot copper block.” The on/off switch is achieved by rapidly pressing a hot copper block against and away from the sensor. To further clarify the temperature changes at the hot and cold ends of the device, we detected the temperature changes at the hot and cold ends in real time through the thermocouple during the test process, and the results are shown in Figure R9. As can be seen, the input temperature difference and output voltage change almost synchronously, which shows a good response from our sensor. The response time and recovery time are ~ 0.5 s and ~ 1 s under different ΔT as shown in Figure R9f.

Fig. R9 Response time of the sensor. (a-e), Voltage response to an input signal of different ΔT from 0.9 K to 17.1 K. The Blue curve represents the temperature signal, and the black curve represents the voltage response of the sensor. (f) Response time as a function of ΔT .

We are a little confused about the reviewer's concern about the sensitivity of $369.6 \mu\text{V K}^{-1}$. As we all know, there is a difference between device sensitivity and material Seebeck coefficient. The sensitivity is a parameter to describe the magnitude of a device's ability to sense temperature differences, while the Seebeck coefficient is a parameter of the thermoelectric performance of the material. Usually, the sensitivity of the sensor will be smaller than the Seebeck coefficient of the materials because of the parasitic thermal resistances. We have mentioned the formula in the manuscript:

$$S = A\alpha = \left(1 - \frac{R_{s1} + R_{o1}}{R_{th} + R_{s1} + R_{o1}}\right)\alpha$$

Where $\alpha = N[(S_p - S_n)]$, S_p and S_n denote the Seebeck coefficients of the Au ($S_p=0$) and Bi_2Te_3 films, respectively, R_{th} denotes the effective thermal resistance, R_{o1} and R_{s1} is the parasitic thermal resistances of the device originating from the PDMS layers, and S is the actual sensitivity of the sensor. So, $S < \alpha$ is easily deduced. Our results are consistent with those reported in the literature, the sensitivity of a 4-pair sensor in the reference¹⁷ is $109.4 \mu\text{V K}^{-1}$ when the Seebeck

coefficient of materials is $42.6 \mu\text{V K}^{-1}$.

We are a little confused about the reviewer's concern about Figure 3c. Figure 3 shows the sensor's performance and Figure 1 shows the performance of Bi_2Te_3 film. It is normal for material flexibility to be slightly stronger than the sensor because the device structure of the electrode, Bonding Ag, and encapsulating material may impact the flexibility. It's pointless to compare the flexibility of Bi_2Te_3 and the sensor together.

In Figure 3e, we have used the normalized sensitivity (sensitivity divided by the number of thermoelectric pairs) rather than sensitivity in our manuscript, which cannot be increased by joining more thermoelectric junctions together. The unit of normalized sensitivity is $\mu\text{V K}^{-1} \text{leg}^{-1}$ in Figure 3e.

Corresponding changes made:

(1) The following descriptions have been added to the revised:

“Fig. 3a shows the sensing performance with five on/off cycles for each ΔT . The on/off switch is achieved by rapidly pressing a hot copper block against and away from the sensor (The input signal of ΔT and the response time of the sensor under ΔT can be seen in Supplementary Figure 12).” (Line 292)

(2) The following figure has been added to Supplementary Fig. 12

7. In Fig 5, the drawback of this kind of device is, resistance measurement needs to be constantly on to measure any pressure change. In real-life applications, this does not seem to be practical.

Response: We thank the reviewer for this valuable comment. We introduced a temperature-pressure sensor based on a bismuth telluride thin film and a 3D spiral structure. The primary innovation lies in the 3D structure enabling planar thin films to match thermal impedance and the bismuth telluride thin film simultaneously exhibiting both piezoresistive and thermoelectric effects. Our research findings demonstrate the feasibility of our concept both theoretically and experimentally. Figure 5

merely illustrates the potential for sensor array measurements, and we conducted measurements using a multi-meter for simplicity. In practical application scenarios, the integration of microcontrollers or other electronic devices would be necessary for comprehensive measurements. However, it is important to note that this aspect is not the primary focus of our research.

8. The homemade dispensing device in Fig S9 appears to be very basic, how can the authors ensure minimal oxidation of the Bi_2Te_3 film? Usually, the process is done in a vacuum. A more elaborate description of Fig S9 setup and details are needed.

Response: We thank the reviewer for this valuable comment. Bi_2Te_3 is relatively stable in air, and even after exposure to air for 5700 hours, the surface oxide layer is only about 2 nanometers thick¹⁸. During the dispensing process, our sliver paste only needs to be held at 413 K for 30 minutes to cure. Therefore, bismuth telluride exhibits minimal oxidation, and its impact on the film's performance is negligible. To demonstrate this, we cycled the Bi_2Te_3 thin film through three heating cycles in ambient air, with each cycle held at 413K for 30 minutes. Figure R10 illustrates the change in the Seebeck coefficient and electrical conductivity after the heat cycles. The result indicates that the performance of the film is nearly unchanged after heating in ambient air.

Figure R11 shows a detailed image of the dispensing device. As can be seen, The Ag paste volume can be tuned by controlling dispensing time using a needle with a diameter of 65 μm . For a tight bond and electrical connection, we choose the 50 ms dispenser time. Figure R11c shows the cross-sectional image during dispensing, which indicates a diameter of $\sim 200 \mu\text{m}$ Ag paster bond zone.

Fig. R10. The electrical performance of Bi_2Te_3 films under heating cycles. Each cycle represents a 30-minute heat period in the air. Error bars represent the measurement uncertainties for Seebeck coefficient ($\sim 3\%$) and electrical conductivity ($\sim 5\%$).

Fig. R11. The image about the dispensing device. (a) Cross-sectional image of silver paste at different dispensing times. (b) Micrograph of the cross-section of a dispensing needle. Scale bar: $50\ \mu\text{m}$. (c) Cross-sectional image during dispensing.

Corresponding changes made:

(1) The following descriptions have been added to the revised methods.

“The deposition mask is designed as shown in Supplementary Fig. 8 and 9. After deposition, the Ag paste was dispensed using a lab-built semiautomatic dispensing machine (Supplementary Fig. 15 left). The Ag paste volume can be tuned by controlling dispensing time using a needle with a diameter of 65 μm . After dispensing, the Ag paste is cured at a 413 K hot plant for 30 minutes.”

(Line 461, revised main text)

(2) Figure R11 has been added in the revised Supplementary Fig. 14.

References:

1. Suo, Z., Ma, E. Y., Gleskova, H. & Wagner, S. Mechanics of rollable and foldable film-on-foil electronics. *Appl. Phys. Lett.* **74**, 1177–1179 (1999).
2. Jin, Q. *et al.* Flexible layer-structured Bi₂Te₃ thermoelectric on a carbon nanotube scaffold. *Nat. Mater.* **18**, 62–68 (2019).
3. Jin, Q. *et al.* Flexible Carbon Nanotube-Epitaxially Grown Nanocrystals for Micro-Thermoelectric Modules. *Adv. Mater.* 2304751 (2023) doi:10.1002/adma.202304751.
4. Huang, B.-L. & Kaviani, M. *Ab initio* and molecular dynamics predictions for electron and phonon transport in bismuth telluride. *Phys. Rev. B* **77**, 125209 (2008).
5. Huang, J. *et al.* Polyimide/POSS nanocomposites: interfacial interaction, thermal properties and mechanical properties. *Polymer* **44**, 4491–4499 (2003).
6. Zhang, L. & Wang, S. Correlation of Materials Property and Performance with Internal Structures Evolvement Revealed by Laboratory X-ray Tomography. *Materials* **11**, 1795 (2018).

7. Kwon, C. *et al.* Multi-Functional and Stretchable Thermoelectric Bi₂Te₃ Fabric for Strain, Pressure, and Temperature-Sensing. *Adv. Funct. Mater.* 2300092 (2023) doi:10.1002/adfm.202300092.
8. Ma, W., Record, M.-C., Tian, J. & Boulet, P. Strain Effects on the Electronic and Thermoelectric Properties of n(PbTe)-m(Bi₂Te₃) System Compounds. *Materials* **14**, 4086 (2021).
9. Hajji, M. *et al.* Strain effects on the electronic and thermoelectric properties of Bi₂Te₃: A first principles study. *Comput. Condens. Matter* **16**, e00299 (2018).
10. Zheng, Z.-H. *et al.* Harvesting waste heat with flexible Bi₂Te₃ thermoelectric thin film. *Nat. Sustain.* (2022) doi:10.1038/s41893-022-01003-6.
11. Pavlova, L. M., Shtern, Yu. I. & Mironov, R. E. Thermal expansion of bismuth telluride. *High Temp.* **49**, 369–379 (2011).
12. Li, H. *et al.* 3D Extruded Graphene Thermoelectric Threads for Self-Powered Oral Health Monitoring. *Small* **19**, 2300908 (2023).
13. Yang, Y., Lin, Z.-H., Hou, T., Zhang, F. & Wang, Z. L. Nanowire-composite based flexible thermoelectric nanogenerators and self-powered temperature sensors. *Nano Res.* **5**, 888–895 (2012).
14. Liu, Y. *et al.* Si/SnSe-Nanorod Heterojunction with Ultrafast Infrared Detection Enabled by Manipulating Photo-Induced Thermoelectric Behavior. *ACS Appl. Mater. Interfaces* **14**, 24557–24564 (2022).
15. Nan, K. *et al.* Compliant and stretchable thermoelectric coils for energy harvesting in miniature flexible devices. *Sci. Adv.* **4**, eaau5849 (2018).

16. Guo, Z. *et al.* Kirigami-Based Stretchable, Deformable, Ultralight Thin-Film Thermoelectric Generator for BodyNET Application. *Adv. Energy Mater.* **n/a**, 2102993 (2021).
17. Zhu, P. *et al.* Flexible 3D Architected Piezo/Thermoelectric Bimodal Tactile Sensor Array for E-Skin Application. *Adv. Energy Mater.* **10**, 2001945 (2020).
18. Bando, H. *et al.* The time-dependent process of oxidation of the surface of Bi_2Te_3 studied by x-ray photoelectron spectroscopy. *J. Phys. Condens. Matter* **12**, 5607–5616 (2000).

REVIEWER COMMENTS

Reviewer #1 (Remarks to the Author):

I think it is ok for publication now

Reviewer #2 (Remarks to the Author):

The authors have provided adequate peer review responses, and it is acceptable to publish this manuscript.

Reviewer #3 (Remarks to the Author):

I have seen the revised version and the response letter. Some improvements have indeed been carried out to the work.

- The literature discussion in the introduction is in a better shape, with adequate comparison to the state of the art.
- Figure R3 adequately addressed my first comment, although part of the optical figure maybe better embed in one of the main figures since this is one of the main novelties of this work.
- Comment 2 and 3 are properly addressed. My fourth comment was intended for the authors to provide more literature comparison of the state of the art (i.e. how is this work better or help to advance the field compared to the latest breakthrough reported in the recent few years?) Figure 3e literature comparison is not so easy to see. It should be improved in terms of clarity.
- Comment 5 has been partially addressed. However, in figure 1B, with every increasing step (corresponding strain), the $dR/R0$ is not perfectly flat, what are the source of errors in the noise?
- Figure R9 has appropriately addressed my comment 6.
- Comment 7 is well addressed.

In view of the good efforts put in by the authors to improve this work, I would be happy to recommend acceptance after the aforementioned minor points are addressed.

RESPONSE TO REVIEWERS' COMMENTS

Reviewer #1 (Remarks to the Author):

I think it is ok for publication now.

Response: We thank the reviewer for the positive reply.

Reviewer #2 (Remarks to the Author):

The authors have provided adequate peer review responses, and it is acceptable to publish this manuscript.

Response: We thank the reviewer for the positive reply.

Reviewer #3 (Remarks to the Author):

I have seen the revised version and the response letter. Some improvements have indeed been carried out to the work.

- The literature discussion in the introduction is in a better shape, with adequate comparison to the state of the art.

Response: We thank the reviewer for this valuable comment.

- Figure R3 adequately addressed my first comment, although part of the optical figure maybe better embed in one of the main figures since this is one of the main novelties of this work.

Response: We thank the reviewer for this valuable comment. We have revised the Fig. 2 and Supplementary Fig 6. The Supplementary Fig. 6a has been deleted. The Supplementary Fig. 6b and c in the original Supplementary file have been moved to Fig. 2 d and e in the revised manuscript.

Corresponding changes made:

(1) The statement has been revised as below:

“The structure features of the sensor and the electrical interconnection are shown in Fig. 2d and Supplementary Fig. 6. As can be seen in Fig. 2d, the sensor contains 3-pair legs. The 3D structure of the spiral is revealed by the X-ray tomography (XRT)⁴⁴, which shows an inverted tower-type spiral structure, as shown in Fig. 2e. Supplementary Fig. 6a further shows the corresponding cross-section of the spiral structure image of Fig. 2e by XRT, where the Bi₂Te₃ film, Ag paste, and Cu foil can be seen. It is worth noting that the Ag paste is tightly attached to the Cu foil, which indicates a good electrical connection. Supplementary Fig. 6b and c shows the diagram of the sensor electrical interconnection. Supplementary Fig. 6c depicts a simplified schematic diagram of the Supplementary Fig. 6b.” (Line 244 – Line 253)

(2) Fig. 2 has been revised, follows shows the revised Fig. 2:

Fig. 2 (a) Magnified image of the part shows the 3D spiral structure. (b) Schematic of the sensor. (c) Sensor folding by hand, demonstrating its excellent flexibility. (d) Optical photograph of the 3-pair leg device. Scale bars, 2 mm. (e) A 3D-spiral structure of one leg in a device constructed by XRT. Scale bars, 250 μm. (f) Schematic for the fabrication of the 3D pressure-temperature sensor.

(3) Supplementary Fig. 6 has been revised as follows:

Supplementary Figure 6 | The spiral structure of the sensor and its electrical connection. (a) The cross-section of the spiral structure imaging by XRT. The red ellipse represents Bi₂Te₃ film, the bright line is Ag paste and the Cu foil is under the Ag paste. Scale bars, 250 μm . (b) Simplified schematic diagram of device electrical connections. (c) The interconnection details of the 3-pair device. The red spiral and blue spiral represent the Au film and the Bi₂Te₃ film, and the black line denotes the Cu foil used as an electrode.

- Comment 2 and 3 are properly addressed. My fourth comment was intended for the authors to provide more literature comparison of the state of the art (i.e. how is this work better or help to advance the field compared to the latest breakthrough reported in the recent few years?) Figure 3e literature comparison is not so easy to see. It should be improved in terms of clarity.

Response: We thank the reviewer for this valuable comment.

Most of the previously reported temperature-pressure dual parameter sensors¹⁻⁹ employ distinct materials to achieve separate sensing functions. In comparison, the pressure and temperature sensing performance are simultaneously realized in our device by using a single flexible bismuth telluride material. Furthermore, a 3D helical structure designed in our study realized an out-of-plane type device, which effectively reducing the response time of the sensors. Additionally, due to the excellent thermoelectric properties of our Bi₂Te₃ film, the normalized temperature sensitivity of our device reaches 123.2 $\mu\text{V K}^{-1} \text{leg}^{-1}$, which is the record-high value.

To better clarify the description, Fig. 3e has been revised as Fig. R1. We compare the temperature sensing performance of our sensor with recently reported bimodal sensors¹⁻⁹. Although many devices with fast response times have been reported in recent years, their improvements in

normalized sensitivity remain limited. Our sensor, with a temperature response times less than 0.5 s and a normalized temperature sensitivity of $123.2 \mu\text{V K}^{-1} \text{leg}^{-1}$, demonstrates superior performance compared to the literature reports, as shown in Fig. R1.

Fig. R1 Temperature sensor performance in this study compared to the literatures¹⁻⁹.

Corresponding changes made:

(1) The following descriptions have been added to the manuscript:

“We compare the temperature sensing performance of our sensors with the recent reports^{9,16,45-47}, as shown in Fig. 3e. Most of the previously reported temperature-pressure dual parameter sensors employ distinct materials to achieve separate sensing functions. In comparison, the pressure and temperature sensing performance are simultaneously realized in our device by using a single flexible bismuth telluride material. Although many devices with fast response times have been reported in recent years, their improvements in normalized sensitivity remain limited. Our sensor exhibits a temperature response times less than 0.5 s, and a normalized temperature sensitivity of $123.2 \mu\text{V K}^{-1} \text{leg}^{-1}$, which demonstrates superior performance compared to the literature reports, as shown in Fig. 3e. (Line 347-354)

(2) Fig. 3e has been revised as follows:

Fig. 3 Sensing performance of the pressure–temperature sensor. (a) Voltage response to temperature difference stimuli from 0.9 to 17.1 K. (b) Output voltage as a function of the sensor temperature gradient. The red line is the fitted curve, exhibiting a sensitivity of $369.6 \mu\text{V K}^{-1}$. The inset shows the time-resolved response of the sensor to temperature stimuli with the red and green zones corresponding to the response and relaxation time, respectively. (c) Resistance variation of the sensor with bending times. (d) Relative resistance changes as a function of the pressure that is applied to the sensor. The inserted image illustrated the test situation. (e) Temperature sensor performance in this study compared to the literature reports^{9,16,45–47}.

- Comment 5 has been partially addressed. However, in figure 1B, with every increasing step (corresponding strain), the dR/R_0 is not perfectly flat, what are the source of errors in the noise?

Response: We thank the reviewer for this valuable comment.

We further investigate the magnitude of the resistance fluctuation. As shown in Fig. 1c, each point was obtained by calculating the average resistance of the samples at each bending radius. The error bars (represent the standard deviation) have been incorporated in revised Fig 1c. The $\Delta R/R_0$ fluctuations in Fig.1b is ~ 0.001 and may stem from the following factors:

1. The relaxation of the polyimide substrate during deformation could cause the resistance fluctuation of the Bi_2Te_3 films.

2. As can be seen in Fig.1a, the conductivity of Bi_2Te_3 film varies with the temperature. Its relative resistance change rate ($\Delta R/R_0$) is about 0.0012 K^{-1} , which means that 1 K temperature fluctuation causes $\Delta R/R_0$ change of 0.0012. So, the resistance fluctuation of our film may derive from the changes in the localized surrounding temperature.

Corresponding changes made:

(1) Fig. 1c has been revised, follows shows the revised Fig. 1.

Fig. 1 Electric and flexible properties of $\text{Bi}_2\text{Te}_3/\text{PI}$ films deposited at 613 K. (a) Temperature-dependent Seebeck coefficient, electrical conductivity, and power factor of the film. (b) Real-time piezoresistance response curve of the film. The numbers indicated by the arrows represent the bending radius. (c) Relative resistance changes of the film as a function of strain. The red line is the fitted curve, whose slope denotes the gauge factor of -9.2. Error bars represent the standard deviation for resistance. (d) Relative electrical resistance as a function of bending radius for the film. R_a and R_0 denote the resistance of the film after bending deformation and the original flat state, respectively. Inset: results of the cyclic bending test under $r = 2.8$ mm for the film. Error bars represent the

measurement uncertainties for resistance (~5%).

- Figure R9 has appropriately addressed my comment 6.

- Comment 7 is well addressed.

In view of the good efforts put in by the authors to improve this work, I would be happy to recommend acceptance after the aforementioned minor points are addressed.

Response: We thank the reviewer for these valuable comments, which have significantly improved the quality of our manuscript. We hope that all the concerns of the reviewer have been well addressed.

References:

1. Zhang, F., Zang, Y., Huang, D., Di, C. & Zhu, D. Flexible and self-powered temperature–pressure dual-parameter sensors using microstructure-frame-supported organic thermoelectric materials. *Nat. Commun.* **6**, 8356 (2015).
2. Gao, F.-L. *et al.* Ti₃C₂T_x MXene-Based Multifunctional Tactile Sensors for Precisely Detecting and Distinguishing Temperature and Pressure Stimuli. *ACS Nano* **17**, 16036–16047 (2023).
3. Zheng, Y., Liu, H., Chen, X., Qiu, Y. & Zhang, K. Wearable thermoelectric-powered textile-based temperature and pressure dual-mode sensor arrays. *Org. Electron.* **106**, 106535 (2022).
4. Zhu, P. *et al.* Flexible 3D Architected Piezo/Thermoelectric Bimodal Tactile Sensor Array for E-Skin Application. *Adv. Energy Mater.* **10**, 2001945 (2020).
5. Wang, J. *et al.* Integrating In-Plane Thermoelectricity and Out-Plane Piezoresistivity for Fully Decoupled Temperature-Pressure Sensing. *Small* 2307800 (2023) doi:10.1002/sml.202307800.
6. Cui, Y. *et al.* Highly Stretchable, Sensitive, and Multifunctional Thermoelectric Fabric for Synergistic-Sensing Systems of Human Signal Monitoring. *Adv. Fiber Mater.* (2023) doi:10.1007/s42765-023-00339-8.
7. Wang, N. *et al.* Pressure-temperature dual-parameter sensors designed by wood-derived thermoelectric composites: Micro-pressure high sensitivity. *Compos. Part B Eng.* **264**, 110928 (2023).
8. Jang, G. N. *et al.* Highly sensitive pressure and temperature sensors fabricated with poly(3-hexylthiophene-2,5-diyl)-coated elastic carbon foam for bio-signal monitoring. *Chem. Eng. J.* **423**, 130197 (2021).

9. Li, M. *et al.* Large-Area, Wearable, Self-Powered Pressure–Temperature Sensor Based on 3D Thermoelectric Spacer Fabric. *ACS Sens.* **5**, 2545–2554 (2020).

REVIEWERS' COMMENTS

Reviewer #3 (Remarks to the Author):

I have gone through the manuscript again carefully. In the latest revised version, the authors have rigorously addressed all my previous comments. Overall the manuscript looks to be in a good shape and high quality. I am happy to recommend acceptance of the current version.

RESPONSE TO REVIEWERS' COMMENTS

Reviewer #3 (Remarks to the Author):

I have gone through the manuscript again carefully. In the latest revised version, the authors have rigorously addressed all my previous comments. Overall the manuscript looks to be in a good shape and high quality. I am happy to recommend acceptance of the current version.

Response: We thank the reviewer for the positive reply.